# Tracing the origins of stratospheric ozone intrusions: direct vs. indirect pathways and their impacts on Central and Eastern China in spring-summer 2019

Kai Meng[1,2], Tianliang Zhao[3], Yongqing Bai[4], Ming Wu[1], Le Cao[3], Xuewei Hou[3], Yuehan Luo[3], Yongcheng Jiang[5]

[1]Hebei Key Laboratory of Meteorology and Ecological Environment, Hebei Provincial Institute of Meteorological Sciences, Shijiazhuang 050021, China
[2]China Meteorological Administration Xiong'an Atmospheric Boundary Layer Key Laboratory, Xiong'an 071800, China
[3]Key Laboratory for Aerosol-Cloud-Precipitation of China Meteorological Administration, Nanjing University of Information Science and Technology, Nanjing 210044, China
[4]Hubei Key Laboratory for Heavy Rain Monitoring and Warning Research, Institute of Heavy Rain, China Meteorological Administration, Wuhan 430205, China
[5]Xiamen Key Laboratory of Strait Meteorology, Xiamen Meteorological Bureau, Xiamen 361012, China

*Correspondence to*: Kai Meng (macka@foxmail.com) and Tianliang Zhao (tlzhao@nuist.edu.cn)

**Abstract:** The impact of stratospheric intrusions (SIs) on Central and Eastern China (CEC) with severe ozone pollution presents a range of uncertainties, underscoring the imperative for sustained research endeavors. In this study, we propose a traceability assessment method that can derive high-resolution critical source areas (CSAs) of SIs by utilizing ambient air quality observations, global ozone (re)analysis data, and customized Lagrangian simulations. This approach enables us to facilitate a meticulous and systematic examination of the impact of direct and indirect SIs on tropospheric and near-surface ozone in six important sub-regions within the CEC during the spring and summer of 2019, as well as the unique circulations driving SIs, from a more refined and targeted tracing perspective. The findings reveal that impacts of indirect intrusions are more efficient at monthly scales, with contributions to tropospheric ozone reaching up to twice the magnitude of direct intrusions. The impacts of direct intrusions are more pronounced at daily scales, primarily occurring in May. In terms of contribution to near-surface ozone, the eastern plains frequently witness ozone exceedance events, with the most substantial impact from SIs observed, (e.g., contributing 15.8% and 16.7% to near-surface ozone in North China and East China, respectively), showcasing a remarkable ability to capture descending lower stratospheric air. In contrast, Loess Plateau and Central China, situated in central and western high-altitude regions, receive more intrusive ozone into the troposphere but exhibit minimal contributions to near-surface ozone. The indirect intrusions that generate the above impact come from 3 to 4 CSAs located thousands of kilometers away, evenly distributed latitudinally within the westerlies between 40° and 70°N (spaced 70 longitudes apart). These CSAs are intricately linked to the evolution of synoptic-scale Rossby waves or subtropical westerly jets, with Western Siberia or Europe and North Africa identified as the most significant CSAs. Conversely, the CSAs for direct intrusions are relatively concentrated, with those influencing the middle troposphere originating from the Tibetan Plateau and those influencing the lower troposphere predominantly located in Mongolia and central Russia, just a few hundred kilometers from the CEC. These sources are associated with typical atmospheric circulations such as the Northeast Cold Vortex and the South Asian High, where the intensity of the intrusion system plays a more crucial role than its frequency of occurrence. This study provides valuable insights for forecasting and mitigating the impact of SIs on ozone pollution in China and contributes to addressing the broader challenges posed by climate change.

**Keywords**: Stratospheric intrusions, Ozone, Stratospheric sources, Central and Eastern China, Atmospheric circulation

# 1 Introduction

High-level ozone not only have adverse effects on crop growth, ecosystems, and human health (Horowitz, 2011; Lu et al., 2020) but also contribute to increased atmospheric oxidation and greenhouse effects (Tai et al., 2014; Tan et al., 2019). Ozone is the sole among the six criteria pollutants stipulated by the Chinese Environmental Air Quality Standards that exhibits substantial background concentrations persisting beyond local control. In addition to emissions from precursor substances and photochemical reactions, ozone in the stratosphere, where 90% of atmospheric ozone is distributed between 10 and 50 km above the ground, plays a significant role in influencing ground-level ozone concentrations and pollution development (Wild, 2007; Bourqui and Trépanier, 2010).

The impact of stratospheric intrusion (SI) is mostly confined to the middle and upper troposphere, where changes in ozone concentrations are significantly influenced by stratospheric-tropospheric transport (Cooper et al., 2005). Recent global warming and the recovery of stratospheric ozone have led to an increased intrusion of stratospheric ozone into the troposphere (Shindell et al., 2006; Fusco and Logan, 2003; Lu et al., 2019). The descent of lower stratospheric air has long been linked to transient increases in ozone at alpine sites, reaching heights in the middle troposphere (Attmannspacher and Hartmannsgruber, 1973; Ambrose et al., 2011; Schuepbach et al., 1999; Stohl et al., 2000; Trickl et al., 2010, 2014). For instance, abnormal changes in surface ozone and meteorological elements were observed on the Tibetan Plateau in summer (Ding and Wang, 2006), and the increases of 15-20 ppbv in ozone were measured at some high-altitude stations in the western United States, constituting more than 20% (Lin et al., 2012). Previous research has led to the widespread belief that SI is only significant at sparsely populated high-elevation sites. However, under specific atmospheric circulation conditions, stratospheric air can be transported downwards further, causing transient increases in near-surface ozone (e.g., Akritidis et al., 2010; Dempsey, 2014; Lefohn et al., 2011; Langford et al., 2015). The measurements during the 2010 California Nexus (CalNex) study (Ryerson et al., 2013) documented the transport of lower stratospheric air to the near-surface layer in the Los Angeles Basin and the nearby Mojave desert (Langford et al., 2012). Pfister et al. (2008) estimated that the impact of SI on near-surface ozone in the United States is nearly equivalent to the influence of local emissions. Akritidis et al. (2010) observed that an SI event elevated nocturnal surface ozone concentrations in Athens, disrupting the typical diurnal ozone variation. Intensive ozone profiling observations reveal that stratospheric ozone mixing in the troposphere is not as extensive as previously believed, and the impact of deep SI on lower tropospheric ozone may be considerably larger than previously thought (Bourqui and Trépanier, 2010; Trickl et al., 2020). Recent applications of high-resolution numerical models and reanalysis data have unveiled more deep SI events than previously unrecognized (Bartusek et al., 2023). The atypical impact of SI on tropospheric, especially near-surface, ozone, coupled with various uncertainties, presents challenges to air pollution control (Akritidis et al., 2018).

Central and Eastern China (CEC) is grappling with a significant ozone pollution issue (Wang et al., 2022). Ozone concentrations in CEC have exhibited a consistent upward trend in recent years (Li et al., 2020). Furthermore, spatial extent of ozone pollution has progressively expanded, emerging as a critical concern for continuous air quality improvement in China. CEC encompasses a vast region, including Northeast China (NE), North China (NC), Loess Plateau (LP), Central China (CC), East China (EC), and South China (SC) (Figure 1e), spanning over 3000 km from north to south. Each sub-region is influenced by distinct atmospheric circulation systems, with distinct topography and vegetable covers. Recent studies have verified that SI affects near-surface ozone in CEC. Zhang et al. (2022), utilizing the GNAQPMS model, assessed the contribution of a SI event to near-surface ozone in NC from August 14 to 18, 2019, estimating it to be between 6% and 20%. Wang et al. (2020) proposed that the combined influence of the South Asian high, subtropical high, and upper trough resulted in a SI event in Nanjing during the summer of 2016, contributing an additional 10 ppbv to surface ozone.

Tropopause folding plays a crucial role in the transport of stratospheric air to the troposphere (Stohl et al., 2003; Browell et al., 1987), primarily occurring near the subtropical jet streams in both the northern and southern hemispheres (Akritidis et al., 2021). The CEC is situated within this SI-influenced control region. In addition to the Brewer-Dobson circulation, substantial air exchange across the tropopause is facilitated by synoptic-scale and mesoscale processes (Danielsen, 1968),

such as blocking highs and cutoff lows (Ancellet et al., 1994), as well as mesoscale convective complexes (Li et al., 2015). Stratospheric air can be transported to the lower troposphere directly through rapid vertical movement (lasting less than 2 days in the troposphere) or indirectly from thousands of kilometers away through a phased tilted mode (resulting in longer tropospheric residence times, such as > 10 days) (Eisele et al., 1999; Meng et al., 2022b; Zanis et al., 1999). Downdrafts associated with typhoons or tropical depressions exhibiting deep convective properties can trigger the direct SI, leading to swift and anomalous changes in near-surface ozone (Venkat Ratnam et al., 2016; Jiang et al., 2015; Zhan et al., 2020). Meng et al. (2022a) estimated that the synergistic impact of typhoons and multi-scale weather systems could cause a 32% increase in surface ozone in the North China Plain. Concerning indirect SI, ozone transport in the troposphere can persist for over 10 days, with different layers of high ozone concentrations influenced by distinct transport sources (Bithell et al., 2000). Utilizing FLEXPART modeling, Akritidis et al. (2010) estimated that the contribution of long-term, long-distance stratospheric ozone intrusion to the western Atlantic could reach up to 6.9%. Our previous study suggested that indirect SI contributed between 5.7% and 18.8% to surface ozone in the North China Plain during the summer of 2019 (Meng et al., 2022b). Unfortunately, much of the existing research on SI's effects on near-surface ozone is confined to limited regions and typical case studies and unilaterally focuses solely on either direct (DSI) or indirect (ISI) stratospheric intrusion.

Our previous study suggested that stratospheric ozone mainly crosses the tropopause in more concentrated regions rather than uniformly entering the troposphere on a larger scale (Meng et al., 2022b). Accurately identifying critical source areas (CSAs), i.e., the zones where stratospheric air masses cross the tropopause and penetrate into the troposphere and even the boundary layer, is closely related to capturing the key atmospheric circulation systems driving SI. This is crucial for accurately predicting and assessing the impact of SI on near-surface ozone. However, few studies have systematically and deeply investigated the stratospheric sources of near-surface ozone in specific regions and their corresponding atmospheric circulation systems. Uncertainties still persist in terms of the contribution rates of SI and regional comparisons. The year 2019 was chosen in this study as a sequel of our previous studies (Meng et.al. 2022a, b) on the influence of stratosphere-to-troposphere transport on summertime surface O3 changes in 2019 and a typhoon-induced SI event in North China Plain in a series of research. This study represents the first attempt to systematically address the impact of SI on six sub-regions within CEC and the regional differences in the impact on tropospheric and near-surface ozone from the perspective of source tracing, distinguishing between DSI and ISI processes. In this study, we introduces a traceability assessment method for SI that integrates ambient air quality observations, global ozone (re)analysis data, and customized Lagrangian simulations. The investigation focuses on the spring and summer of 2019, with the primary objective of systematically identifying high-resolution sources of both DSI and ISI that impact ozone levels in the middle (3-7 km) and lower (<2 km) troposphere across the CEC region. Furthermore, the study aims to identify key circulation systems linked to the intrusion source areas, enhancing the capacity to predict and respond to the impact of SI events on ozone pollution in China.

## 2 Data and Methods

### 2.1 Data

In this study, we utilized the meteorological fields, including geopotential height, potential vorticity, ozone mixing ratio, and wind, from the 0.75° ERA-Interim reanalysis data available from the European Center for Medium-Range Weather Forecasts (ECMWF) (https://cds.climate.copernicus.eu/cdsapp#!/dataset/reanalysis-era-interim?tab=form). ERA-Interim data are solely used for large-scale atmospheric circulation analysis, as detailed in Section 3.5. SIs are primarily controlled by large-scale circulation, and our analysis using ERA-Interim data at 0.75° resolution effectively addresses the scientific issue. The daily surface ozone concentrations were obtained from the operational observations of the Ministry of Ecology and Environment of China (https://air.cnemc.cn:18007/).

The EAC4-CAMS (https://ads.atmosphere.copernicus.eu/cdsapp#!/dataset/cams-global-reanalysis-eac4) and the global atmospheric chemistry model CAM-Chem (https://www.acom.ucar.edu/cam-chem/cam-chem.shtml) both provide stratospheric ozone tracer products ($O_3S$) suitable for analyzing stratosphere-troposphere transport events (Akritidis et al., 2022). EAC4, the fourth-generation global meteorological and atmospheric composition reanalysis dataset released by ECMWF's Copernicus Atmosphere Monitoring Service (CAMS), includes three-dimensional fields of meteorological elements, chemical species, and aerosols (Inness et al., 2019), with a horizontal resolution of 0.75°×0.75° and 60 vertical layers. CAM-Chem, derived from the coupling of MOZART-3 stratospheric chemistry and the MOZART-4 mechanism to the Community Atmosphere Model (Tilmes et al., 2016), features a horizontal resolution of 0.9° × 1.25° and 56 vertical layers.

The Aura Ozone Monitoring Instrument (OMI) Level-2 Ozone Profile data product OMO3PR is available from NASA's Goddard Earth Sciences Data and Information Services Center (GES DISC) (https://disc.gsfc.nasa.gov/datasets/OMO3PR_003/summary). The OMO3PR Level-2 ozone profile product, at a pixel resolution of 13x48 km (at nadir), is based on an optimal estimation algorithm with climatological ozone profiles as a priori information. The ozone profile is represented in terms of layer-column ozone in Dobson Units (DU) for an 18-layer atmosphere, with layers nominally bounded by pressure levels: surface pressure, 700, 500, 300, 200, 150, 100, 70, 50, 30, 20, 10, 7, 5, 3, 2, 1, 0.5, and 0.3 hPa. In this study, DU units are converted to parts per billion by volume (ppbv) using the formula: $(1.2672 \cdot N_i)/DP_i \cdot 1000$, where $N_i$ represents the layer-column ozone in DU and $DP_i$ is the pressure difference between the top and bottom of each layer in hPa.

The three-dimensional ozone climate data were obtained from the Trajectory-mapped Ozonesonde dataset for the Stratosphere and Troposphere (TOST). This dataset assimilated more than 160 ozone soundings from the World Ozone and Ultraviolet Radiation Data Centre (WOUDC) using the Lagrangian method (Liu et al., 2013a). Compared to satellite ozone observations, the TOST ozone data can effectively capture the temporal and spatial variations of global stratospheric and tropospheric ozone levels, as well as the changes in ozone gradients near the tropopause (Liu et al., 2013b). The TOST dataset has a horizontal resolution of 5° × 5° and a vertical resolution of 1 km, covering a range of ground to 26 km in altitude. We employed this data primarily for background climatic concentrations in the lower stratosphere when calculating the CSAII index.

The three-dimensional ozone climate data were obtained from the Trajectory-mapped Ozonesonde dataset for the Stratosphere and Troposphere (TOST). This dataset assimilated more than 160 ozone soundings from the World Ozone and Ultraviolet Radiation Data Centre (WOUDC) using the Lagrangian method (Liu et al., 2013a). Compared to satellite ozone observations, the TOST ozone data can effectively capture the temporal and spatial variations of global stratospheric and tropospheric ozone levels, as well as the changes in ozone gradients near the tropopause (Liu et al., 2013b). The TOST dataset has a horizontal resolution of 5° × 5° and a vertical resolution of 1 km, covering a range of ground to 26 km in altitude. We employed this data primarily for background climatic concentrations in the lower stratosphere when calculating the CSAII index.

## 2.2 Model configuration

In this study, we used FLEXPART v9.3 (https://www.flexpart.eu/downloads, Stohl et al., 2005, 1998), which was driven by meteorological fields from the Global Forecast System (GFS) (Kalnay et al., 1990; Kanamitsu, 1989) provided by the U.S. National Weather Service (NWS) (http://nomads.ncep.noaa.gov/). The FLEXPART model, which was developed by the Norwegian Institute for Air Research (NILU), is a Lagrangian particle diffusion model that accurately models the transport, diffusion, dry and wet deposition, aging, and decay of tracers in the atmosphere by calculating the trajectories of released particles (Brioude et al., 2013). To improve the simulation capability of wet convection and turbulence processes, the Emanuel convection parameterization scheme (Emanuel and Zivković-Rothman, 1999) and the Hanna boundary layer

parameterization scheme (Hanna, 1982) were incorporated into the FLEXPART model. The FLEXPART model has been extensively used in individual cases and long-term experimental studies of air pollutant transport (Forster et al., 2004; Stohl and Trickl, 1999; Stohl and Wotawa., 1998) and can accurately identify the stratospheric-tropospheric transport (Bourqui, 2004).

We conducted a daily rolling simulation with 15-day forward trajectories of over 500,000 air particles released from the stratosphere over the Eurasian region (-20∘W–180∘E) between May 1 and August 31, 2019. In our FLEXPART simulation, we employed domain-filling technology (Chen et al., 2012; Drumond et al., 2010) to simulate the stratospheric ozone tracer, ensuring that only air masses released from the stratosphere were considered. During the initial stages of simulation, particles in the troposphere were filtered out, while stratospheric particles were assigned mass according to the

formula: $M_{O_3} = M_{air} \cdot P \cdot C \cdot 48/29$, where $M_{air}$ denotes the air mass of the particle, P is the potential vorticity, C=60×10$^{-9}$ pvu$^{-1}$ is the ozone/PV scaling factor (Stohl et al., 2000, 2005), and the factor 48/29 is applied to convert volume to mass mixing ratio. Throughout the integration, particles were advected from the stratosphere into the troposphere driven by GFS wind fields. In mid-latitudes, the potential vorticity (PV) is commonly regarded as an indicator of the dynamical tropopause (Bellevue et al., 2007). PV values ranging from 1.0 or 1.6 PVU (Stohl et al., 2000) to 3.5 PVU (Hoerling et al., 1991) are

typically used (Akritidis et al., 2016, 2018, 2021; Hoskins and Berrisford, 1988; Skerlak et al., 2014; Sprenger et al., 2003; Stohl et al., 2003). Gerasopoulos et al. (2006), for instance, used a PV greater than 1.5 PVU in backward trajectories to track air masses of stratospheric origin. In this study, we defined the dynamic tropopause as a PV of 1.5 PVU and stratospheric air as a PV greater than 1.5 PVU. The model output included three-dimensional ozone concentrations and position information of particles at 3-hour intervals, a horizontal resolution of 0.3∘ × 0.3∘, and 17 vertical layers spaced 500 m apart.

The contributions (see Sect. 3.2) presented in this study originate from ozone concentration calculations at various altitudes within the troposphere obtained from our FLEXPART simulation. The calculation of concentration within each model grid is achieved by sampling tracer mass fractions of all particles within the grid cell and dividing by the grid cell volume:

$$C = 1 \Big/ V \cdot \sum_{i=1}^{N} (m_i f_i)$$

Here, V represents the volume of the grid cell, $m_i$ denotes the mass of particle i, N is the total number of particles, and

$f_i$ represents the mass fraction contributed by particle i to the respective grid cell. The mass fraction $f_i$ is computed using a uniform kernel with grid distances $\Delta x$ and $\Delta y$ in the longitude-latitude output grid.

**2.3 CSA identification and CSA impact index**

We employed the three-dimensional trajectory information obtained from the FLEXPART simulation to identify particles that reached the middle and lower troposphere within each sub-region of CEC, either directly or indirectly from the

stratosphere. Unique IDs were then assigned to these trajectories/particles, enabling seamless tracking during subsequent analyses. By assimilating the meteorological field with the trajectory information and applying the distance weighting method, we determined the spatial location of each trajectory as it crossed the tropopause. The PV values, interpolated from meteorological reanalysis data, were used to determine the dynamical tropopause. Specifically, we adopted latitude-variable PV thresholds with 1.5 PVU north of 30°N and 3.0 PVU south of 30°N to define the dynamical tropopause and assesses the

stratospheric origin of air masses over the study regions in China. This approach constrained ozone to originate exclusively from the stratosphere, crossing the tropopause into the troposphere. This methodology enabled us to identify the critical source areas (CSAs) of SI, which are the zones where stratospheric air masses penetrate into the troposphere by crossing the tropopause. They are characterized by a high proportion of gridded particles/trajectories information within these spatial

locations.

Subsequently, we defined a CSA impact index (CSAII). Considering that the amount of stratospheric particles received largely depend on the size of sub-regions, CSAII was defined by dividing the number of trajectories passing through CSAs by the respective area of each CEC sub-region and then multiplying it by the monthly mean ozone concentrations in the lower stratosphere obtained from the TOST dataset. The distribution of CSAII characterizes the geographical spread of CSAs and facilitates an assessment of the impact of each CSA on the middle and lower tropospheric ozone.

## 210 2.4 Validation of the Lagrangian simulations

Validating the stratospheric trajectories and associated vertical profiles of ozone is crucial for understanding the key features of stratospheric intrusion (SI) events. Given the challenges in observing stratospheric ozone in the troposphere, assessing stratospheric intrusions and their vertical structure is inherently difficult. We validated the reliability of our FLEXPART simulations for stratospheric trajectories in the troposphere by comparing simulated daily SI ozone with stratospheric ozone 215 tracers from two reanalysis datasets of atmospheric compositions, as well as the satellite remote sensing of tropospheric ozone.

### 2.4.1 Comparisons with stratospheric ozone tracers

Although continental-scale tracer experiments offer unique opportunities to test Lagrangian simulations, few such experiments have been conducted (Stohl et al., 1998). Here, we employed two reanalysis datasets (EAC4 and CAM) 220 containing $O_3S$ as alternatives to analyze correlations with SI ozone from May to August 2019 simulated across six sub-regions, as depicted in the boxes of Figure 3e. EAC4 and CAM have horizontal resolutions of 0.75°×0.75° and 0.9°×1.25°, respectively, and are capable of resolving daily SI processes. We performed two cross-validations (FLEXPART vs EAC4 and FLEXPART vs CAM) of stratospheric ozone intrusion and transport in the troposphere. Figure 1 shows strong correlations between SI ozone and the two $O_3S$ datasets, with correlation coefficients in most sub-regions exceeding 0.7 and meeting the 225 confidence threshold of 99.9%. Although the magnitude of $O_3S$ in EAC4 is slightly lower, the scatter patterns of stratospheric ozone tracers from both reanalysis datasets are similar to SI ozone (Fig.1), exhibiting a stronger stratospheric signal in northern sub-regions and a weaker impact on low-latitude regions.

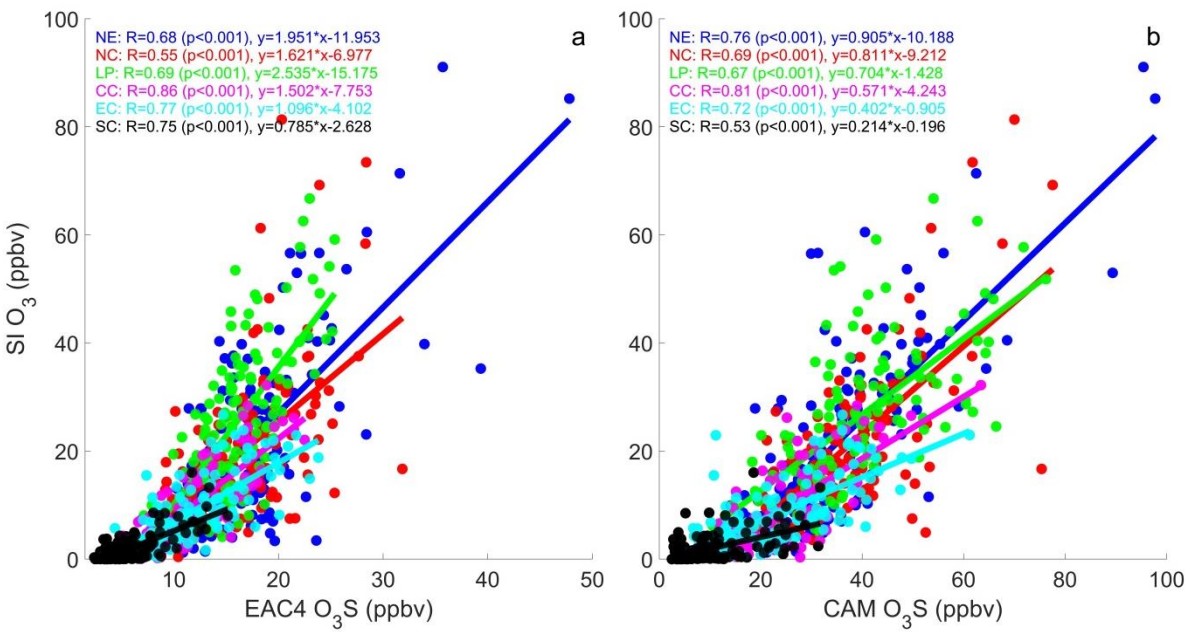

**Figure 1: Scatter plots comparing FLEXPART SI ozone with a) EAC4 and b) CAM stratospheric ozone tracers ($O_3S$) from May to**
230 **August 2019, with six different colors representing the six sub-regions.**

### 2.4.2 Comparison with vertical structure of OMI tropospheric ozone

We examined the spatial correspondence between SI ozone and actual tropospheric ozone from the OMI remote sensing data (Figure 2). We selected OMI's measurement altitudes of 400, 600, and 850 hPa to represent the upper, middle, and lower troposphere, respectively. The vertical structure of OMI tropospheric ozone shows a distinct decrease and finer details in the troposphere with decreasing altitudes, with SI ozone exhibiting similar patterns. Horizontally, OMI ozone in the upper troposphere shows a typical north-high, south-low distribution, consistent with SI ozone (Figure 2a). In the middle troposphere, the distribution generally maintains a north-south pattern, with high ozone values extending eastward from East Asia to the western Pacific, and SI ozone responds similarly (Figure 2b). In the lower troposphere, OMI ozone shows an east-high, west-low distribution, with SI ozone displaying comparable characteristics, such as high values between 30°N and 40°N extending eastward from mainland China, forming a high ozone transport belt similar to the middle troposphere (Figure 2c). The vertical variations of OMI and SI ozone exhibit similar spatial patterns over CEC in the troposphere, while the simulated SI ozone concentrations are lower than OMI ozone, highlighting reasonable relationships between stratospheric impact and total ozone (SI + tropospheric production) in the troposphere and underscoring the credibility of our simulation methodology.

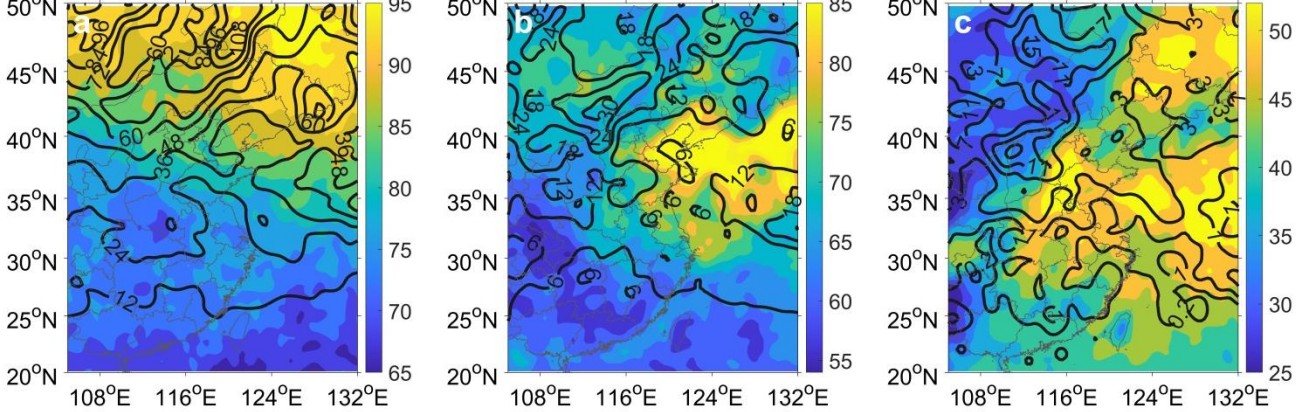

**Figure 2: Spatial correspondence between SI ozone (ppbv, black contours) and actual tropospheric ozone from the OMI remote sensing data (ppbv, color contours) from May to August 2019 at altitudes of a) 400, b) 600, and c) 850 hPa.**

## 3 Results and Discussion

### 3.1 Temporal and spatial variations of tropospheric ozone over CEC

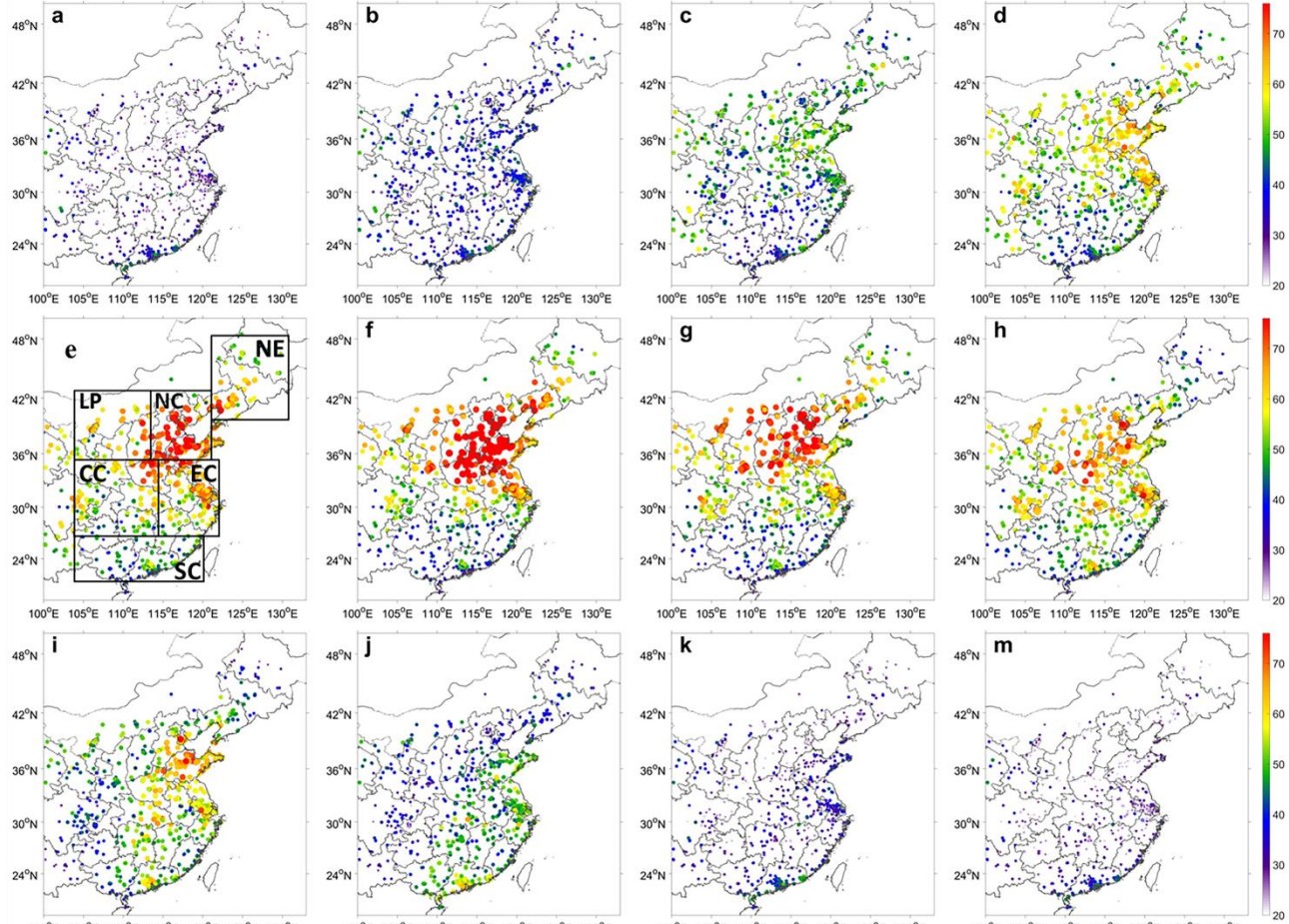

**Figure 3: Distributions of the observed near-surface O₃ concentrations (ppbv) over CEC for (a-l) January to December during 2015-2019 with the sub-regions NE (NorthEast China); NC (North China); LP (Loess Plateau); CC (Central China); EC (East China); SC (Southern China).**

Figure 3 shows the monthly mean of near-surface ozone concentrations in CEC from 2015 to 2019. The results indicated that the near-surface ozone concentrations in CEC gradually increase from January, with a peak between May and July, followed by the gradual declines. Concerning spatial distribution, NC (North China) had the highest level of ozone concentrations, followed by the LP (Loess Plateau) and EC (Eastern China), while CC (Central China) and NE (NorthEast) exhibited relatively lower concentrations, and SC (Southern China) had the lowest. Although varying seasonal patterns of ozone existed in different sub-regions, the spatial distribution highlighted the common characteristic of late spring-summer ozone peaks for CEC.

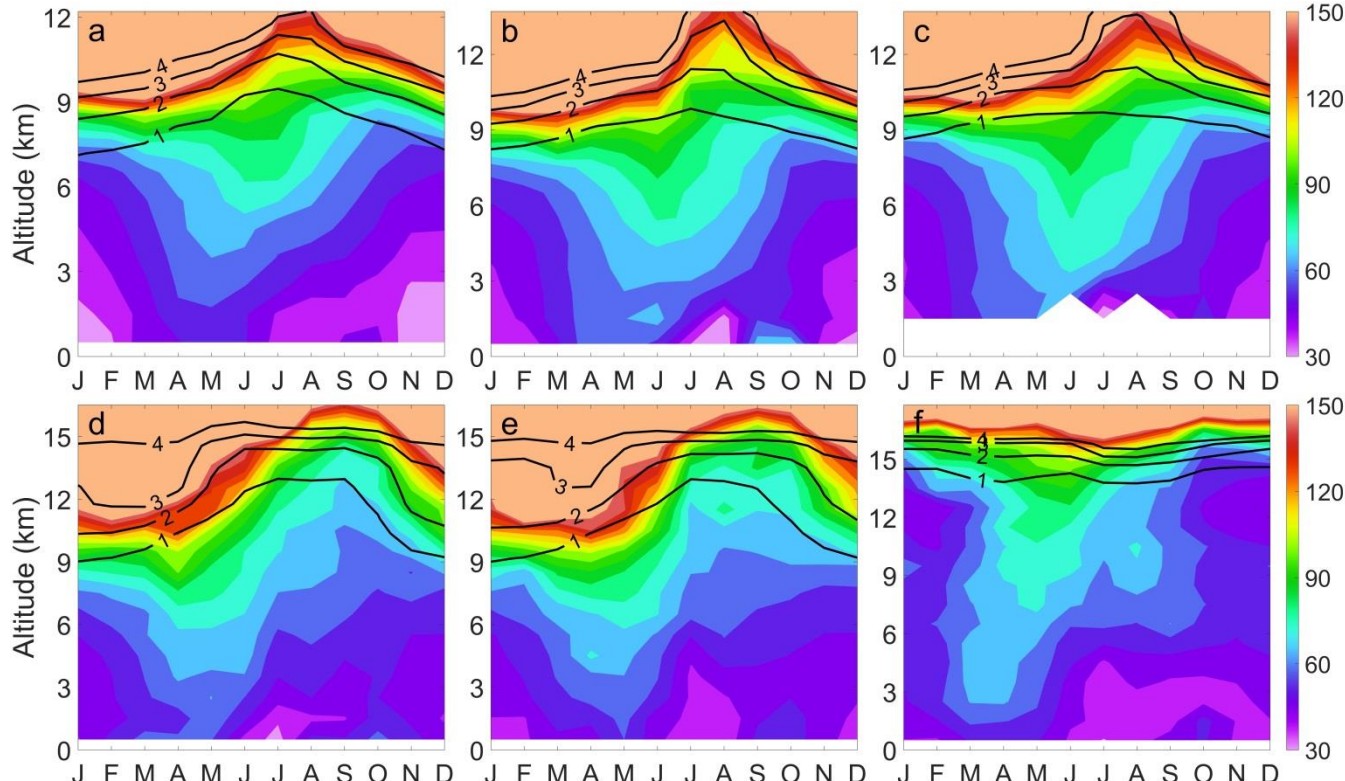

**Figure 4: Monthly variations of vertical O₃ concentrations (ppbv) of TOST averaged during 1990-2019 for a) NE, b) NC, c) LP, d) CC, e) EC and f) SC. The black lines are PV isolines of 1-4 PVU.**

The monthly variations in tropospheric ozone over six sub-regions (Fig. 4) reveals a consistent seasonality of increasing ozone levels from winter to spring, reaching peak from late spring to summer, and subsequently decreasing from autumn to winter. During spring and summer, all six sub-regions exhibit elevated ozone levels spanning from the stratosphere to the lower troposphere, mirroring the pattern of the spring-summer ozone maximum observed at ground level. This recurring spring-summer maximum phenomenon is consistent with the findings of extensive analyses of near-surface and tropospheric ozone observations conducted over the long term (Feister and Warmbt, 1987; Staehelin et al., 1994; Ancellet and Beekmann, 1997), and it is identified in other regions around the world. Beekmann et al. (1994) proposed that the tropospheric spring-summer ozone maximum is a result of SI and regional photochemical productions. However, a consensus on the precise mechanism responsible for this tropospheric spring-summer maximum has yet to be reached. The principal objective of this study is to investigate the different impacts of SIs on the spring-summer ozone in these six sub-regions and to identify the critical sources of this impact.

**3.2 Influence of direct and indirect transports on tropospheric ozone over CEC**

Previous studies (Eisele et al.; 1999; Meng et al., 2022b; and Zanis et al., 1999) proposed that stratospheric air can be transported to the lower troposphere directly through rapid vertical movement (lasting less than 2 days in the troposphere) or indirectly from thousands of kilometers away through a phased tilted mode (resulting in longer tropospheric residence times, such as > 10 days). In this study, we have expanded upon these concepts to distinguish between direct and indirect stratospheric intrusions, where direct intrusions involve stratospheric air from specific sources reaching mid or lower troposphere regions within approximately two days, while indirect intrusions extend this period to around ten days. Notably, both mid and lower tropospheric regions are considered endpoints for stratospheric air transport in our definitions, with transit times set at two and ten days for these sub-regions, respectively. Furthermore, our definitions emphasize the concept about sources of SI. Accordingly, we employed 2-day and 10-day forward trajectories originating from stratosphere simulated by the FLEXPART model to represent direct and indirect transports of stratospheric ozone, respectively. The contributions presented here are derived from ozone calculations at different altitudes in the troposphere obtained from our

FLEXPART simulation. Subsequently, we conducted an analysis of the impact of these two types of transports on ozone concentrations in the middle and lower troposphere within each sub-region of the CEC from May to August in 2019.

### 3.2.1 Direct transport

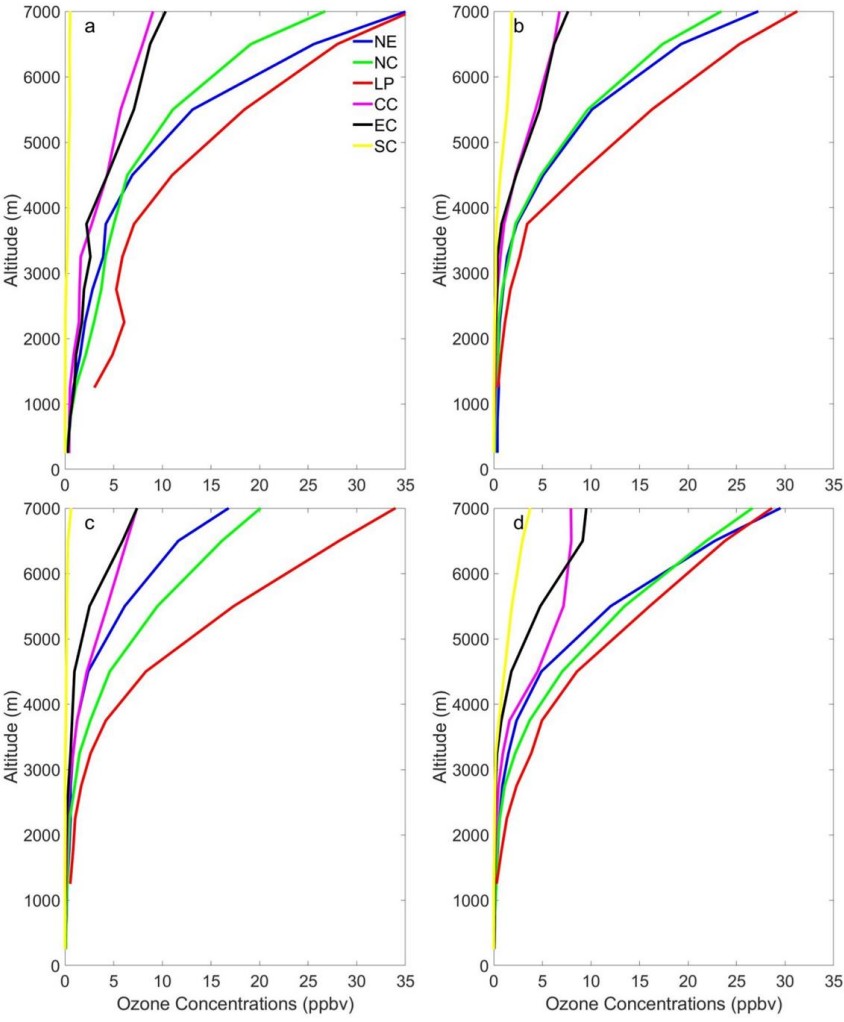

**Figure 5: Vertical profiles of contributions of stratospheric ozone (ppbv) directly transported to the troposphere over six sub-regions of CEC from May to August in 2019: a) May, b) June, c) July, and d) August. The altitude is above sea level (a.s.l.).**

Figure 5 depicts the vertical contributions of stratospheric ozone directly transported to the middle and lower troposphere over CEC sub-regions by 2-day trajectories from May to August. The contribution of direct transport to the middle tropospheric ozone surpasses that in the lower troposphere, with the three northern sub-regions LP, NC, and NE exhibiting higher contributions to the tropospheric ozone levels. Below 2 km, the average contributions experience sharp declines, dropping drastically to less than 2 ppbv. In contrast to the three northern sub-regions closer to the subtropical westerly jet, the southern sub-regions EC and CC show less contribution to tropospheric ozone from the upper atmosphere to the boundary layer. Particularly, the vertical distribution of ozone contribution in the southernmost sub-region SC was relatively uniform, displaying the lowest stratospheric influences on ozone levels in the troposphere among all six sub-regions.

The magnitude of stratospheric ozone transported to the lower troposphere remains comparable among the six sub-regions during the East Asian summer monsoon from May to August. Below 2 km, the received stratospheric ozone is exceedingly low, occasionally punctuated by spikes of high ozone from direct (DSI) stratospheric intrusions observed in NE, NC, LP, CC, and EC. For instance, in May, an enhanced DSI event transpired, resulting in DSI concentrations reaching 5.2, 13.1, 22.4, 6.4, and 11 ppbv in the lower troposphere of the respective sub-regions. In contrast, SC did not experience a significant DSI event, and the maximum DSI concentration in the lower troposphere of SC was merely 0.4 ppbv, reflecting the lowest DSI

contribution of stratospheric ozone to southern China.

### 3.2.2 Indirect transport

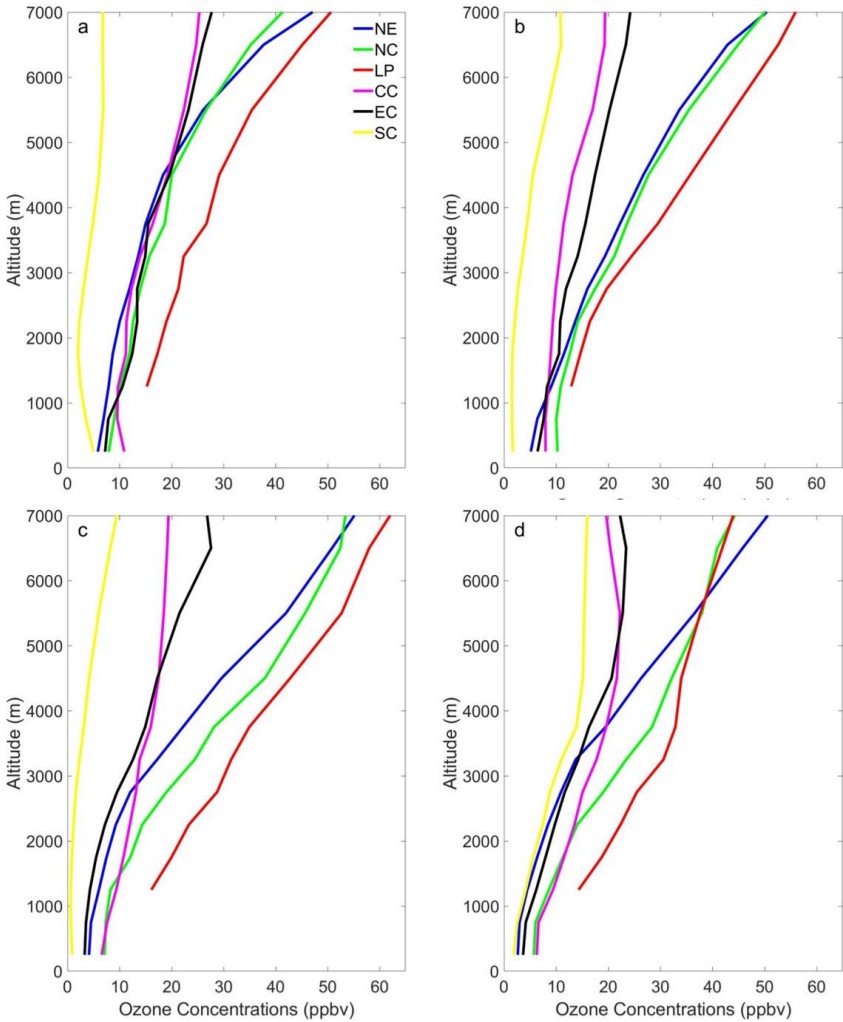

**Figure 6: Same as Fig. 5 but for indirect transport.**

The vertical profiles of ozone resulting from indirect transport with 10-day trajectories depict the elevated ozone levels in the middle troposphere gradually declining in the lower troposphere (Fig.6), sharing similar characteristics with direct transport. Nonetheless, tropospheric ozone resulting from indirect transport notably surpasses that of direct transport. Ozone transported indirectly to the middle troposphere registers twice the magnitude of the direct transport, and ozone conveyed by indirect means to the lower troposphere also surpasses that of direct transport, reaching levels of up to 5 ppbv or even exceeding 10 ppbv. This indicates that ozone transport by indirect means over monthly scales is more efficient.

The analysis above indicates that, whether through direct or indirect transport, SI notably contributes to heightened ozone concentrations in the middle troposphere of the three northern sub-regions (LP, NE, and NC), while its impact on tropospheric ozone is minimal in the southern region, especially in SC. Oltmans et al. (2004) proposed that the elevated ozone in the SC troposphere during spring and summer primarily originates from long-distance transport of biomass burning and high-altitude lightning, rather than contributions from stratosphere-troposphere transport.

### 3.3 Contributions of DSI and ISI to near-surface ozone

We identified DSI and ISI events occurring from May to August in 2019 based on the thresholds of near-surface SI ozone originating from the stratosphere derived from 2-day and 10-day FLEXPART trajectories, reaching 1.5 ppbv and 10 ppbv, respectively. These thresholds were determined using the 75th percentile of SI ozone concentrations. Days that did not meet

these criteria were considered intrusion-free periods.

### 3.3.1 Comparisons between model simulations and near-surface measurements

**Table 1 Agreement (in number of events) between model simulations and measured SI events in six sub-regions, along with the corresponding hit rates of simulations.**

| Sub-regions | Measurements | Simulations | Hit rates (%) |
|---|---|---|---|
| NE | 7 | 5 | 71 |
| NC | 7 | 6 | 86 |
| LP | 32 | 17 | 53 |
| CC | 15 | 12 | 80 |
| EC | 9 | 8 | 89 |
| SC | 3 | 1 | 33 |

Detailed validations of simulations against surface observations are essentially important. We validated our SI assessment method using ground-based records. Chen et al. (2024) determined that significant deviations in ozone and CO from their normal values indicate an SI event if ozone concentrations at the start hour of SI exceed the seasonal mean at noon, while CO concentrations decline below their seasonal mean during the SI. Cui et al. (2009) identified SI events by observing surface ozone exceeding 10% of the 10-day running mean for at least six consecutive hours, with relative humidity below 50%. Considering the variability of moisture conditions of air parcels from higher altitudes to the surface, relative humidity is less conservative than CO (Dreessen, 2019). Thus, both high ozone and low CO values can signal a stratospheric influence (Bonasoni et al., 2000). Based on these methods, we propose a ground-based SI event identification criterion to validate the reliability of model simulations in each sub-region. The criterion involves observing near-surface ozone exceeding 10% of the 10-day running mean for at least six consecutive hours while CO concentrations are lower than the 10-day running mean. Additionally, our method considers the regional coverage for an SI event affecting near-surface air quality, which requires at least two-thirds of the total stations in a sub-region (113 to 302 stations) to meet the assessment criteria. Since the observational method cannot differentiate between types of SI, if either ISI or DSI events identified by the simulation match a measured SI event, the model is considered to have accurately hit the measured SI event.

The number of measured SI events, the number of times these events were hit by model simulations, and the hit rates are listed in Table 1. It is worth noting that the actual number of SI events simulated in this study is greater than the number of measured events hit by the model, due to the application of the stringent 6-hour ozone increase and CO decrease criterion from the literature, which results in fewer SI events. Except for the LP and CC sub-regions, the number of measured SI events affecting near-surface ozone in other sub-regions during spring and summer 2019 is fewer than 10, consistent with Chen et al. (2024), who reported 8-12 SI events annually in China. The LP sub-region, situated on a plateau, experienced 32 SI events, comparable to the over 100 SI events reported by Cui et al. (2009) at a higher altitude site (3585 m above sea level) in Sweden. The frequency of SI events from our method aligns with these studies. Our simulation results indicate a high hit rate for SI events across all sub-regions, except for SC. The low hit rate in SC is not attributable to small PV values but rather to minimal SI impacts on this sub-region during the spring and summer of 2019, with SI not been the primary driver of elevated ozone (Oltmans et al., 2004). Therefore, the hit rate for SC is less significant.

### 3.3.2 Contributions of SI events to near-surface ozone

The frequencies and durations of DSI events in all CEC sub-regions were notably lower than those of ISI, primarily

occurring in May, consistent with the study of SI events in the Italian northern Apennines by Cristofanelli et al. (2006). The high-altitude sub-region LP experienced DSI events exclusively in July and August with the highest number of DSI days. Conversely, the SC region exhibited no significant DSI events. In contrast, ISI occurrences spanned a more extended timeframe, transpiring from May to August for the sub-regions NC, LP, CC and EC. Compared to DSI, ISI demonstrated greater temporal continuity, with two periods lasting more than a week from late May to early June and late June to early July.

To assess the impact of SI on near-surface ozone, we extracted near-surface ozone observations during daytime and nighttime for DSI, ISI, and intrusion-free periods (Fig. 7). It is important to note that the goal here is to compare the average ozone concentrations over complete daytime and nighttime periods, which differs from the aforementioned criteria of a continuous six-hour period. Although ozone levels exhibited variations across six sub-regions, the overall ozone concentrations during ISI periods were the highest (red frames in Fig. 7), indicating a significant role of ISI in the substantial increase of near-surface ozone. Notably, near-surface ozone concentrations during DSI events (green frames in Fig. 7) were generally lower than those during both ISI and intrusion-free periods (blue frames in Fig.7), particularly during the daytime. This implies that under specific conditions, near-surface ozone concentrations during SI events may be lower than those recorded under the influence of photochemical production episodes (Cristofanelli et al., 2006). This disparity may be attributed to the comparatively low concentrations transported by DSI, which did not effectively overcompensate for near-surface ozone. The weakening of the entrainment process by strong tropospheric subsidence may be another contributing factor, limiting the influence of stratospheric ozone transport on the augmentation of near-surface ozone (Kalabokas et al., 2020).

The average contributions of SI to near-surface ozone were quantified. The impact of SI on near-surface ozone is not a simple linear problem, given the complex interactions of various factors such as turbulent mixing and underlying surface characteristics when stratospheric ozone enters the atmospheric boundary layer. Therefore, assessing the contribution of SIs to near-surface ozone is not straightforward by merely dividing SI ozone concentrations by surface ozone observations. To minimize confounding factors, we chose the nighttime period to calculate the contributions of SI to near-surface ozone. In this study, we employed a method involving the subtraction of nighttime surface ozone observed on SI days from the average concentrations during intrusion-free periods. The resulting difference is then normalized by the average concentrations during the intrusion-free periods to assess the contribution.

For ISI, the contributions, ranked from highest to lowest, are as follows: SC (38.3%), NE (25.5%), EC (16.7%), NC (15.8%), CC (1.4%), and LP (-7.5%). It is important to realize correctly that the negative percentage for the sub-region LP does not imply that SI causes a decrease in near-surface ozone. Instead, it indicates that near-surface ozone levels in LP during intrusion periods, on average, are lower than during intrusion-free periods (see Figure 7), despite the fact that ozone originating from the stratosphere and reaching LP is the highest. Trickl et al. (2023) scrutinized observations at the Alpine station Zugspitze and found that SI occurrences do not consistently lead to an increase in ground-level ozone. This may be due to the low efficiency of certain regions' atmospheric boundary layers in capturing descending stratospheric air to supplement near-surface ozone. It is also noteworthy that ISI occurrences for SC are limited, leading to high contributions but with notable uncertainties. Similarly, due to limited DSI occurrences for all six sub-regions, the DSI contributions are not computed. The outlined contributions highlight several issues. The most substantial contribution from ISI is observed in the eastern sub-regions (NE, NC, EC, etc.), while contributions to LP and CC, located in the central and western regions, are minimal. Previous studies have fostered the widespread belief that SI is only significant at high-elevation sites (Attmannspacher and Hartmannsgruber, 1973; Ambrose et al., 2011; Stohl et al., 2000; Trickl et al., 2010, 2014). However, our study shows that SI may play an important role in modulating near-surface ozone in the eastern plains compared to the central and western high-altitude regions of the country. These discrepancies in the contribution of SIs to ground-level ozone may be linked to variations in atmospheric boundary layer mixing and chemistry processes (Hu et al., 2013; Caputi et al.,

2019; Makar et al., 2017).

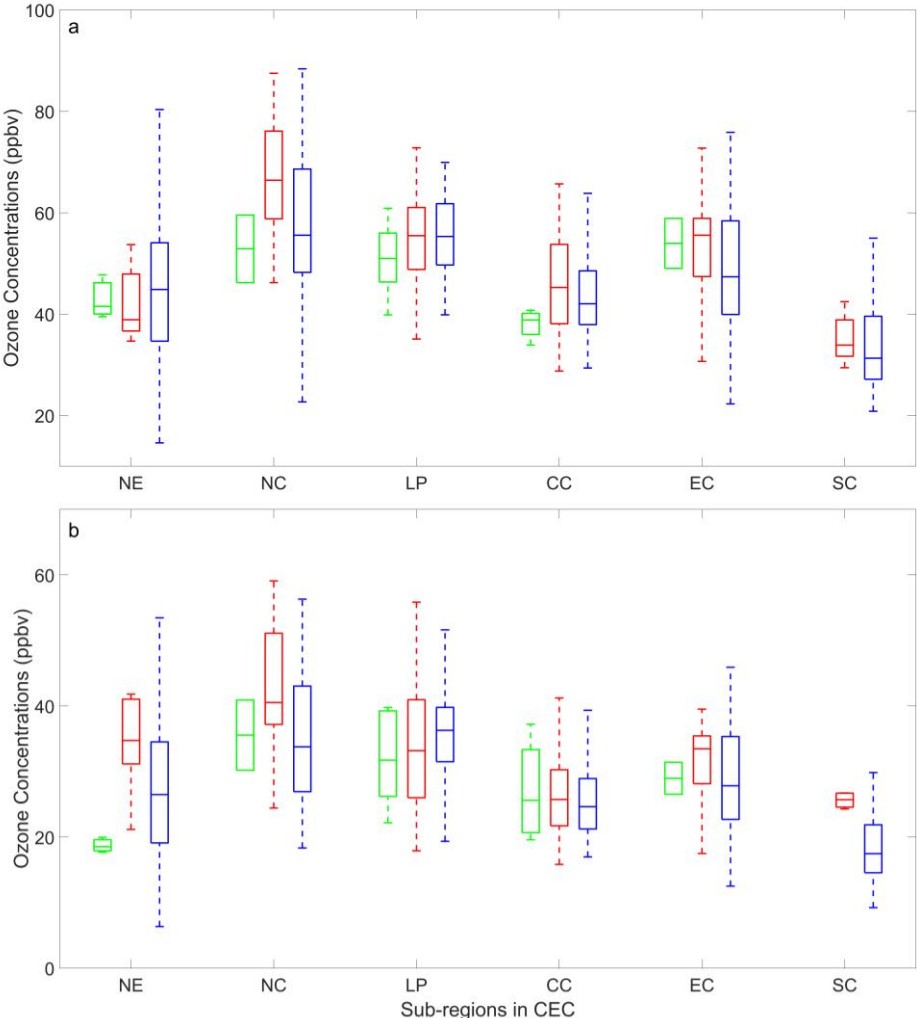

Figure 7: Boxplots of near-surface ozone concentrations during daytime (a) and nighttime (b) in six sub- regions for the periods of DSI (green) and IST (red) and intrusion-free conditions (blue). Boxes indicate the distribution of near-surface ozone concentrations between the 25th and 75th percentiles, with the horizontal lines inside the boxes denoting the median O3 concentrations. 9 direct SI samples affecting LP, and 2-4 samples affecting the remaining five sub-regions over CEC, are used to perform the statistical analysis.

### 3.4 Distribution of CSAs and their impacts

This section delves into the Critical Source Areas of SIs, which denote the spatial locations where stratospheric ozone traverses the tropopause, exerting a notable influence on tropospheric ozone within each sub-region of CEC. The CSA Impact Index was used to analyze the SI sources. The distribution of this index not only characterizes the geographical spread of CSAs but also facilitates an assessment of the extent to which each CSA impacts the middle and lower tropospheric ozone levels.

#### 3.4.1 CSAs for ISI

Figure 8 illustrates the spatial distribution of CSAII of ISI for six sub-regions of CEC. The solid lines denote CSAs influencing the middle troposphere (CSAm), while the color shading represents CSAs impacting the lower troposphere (CSAl). The spatial pattern reveals discernible distinctions in the stratospheric sources affecting the middle and lower troposphere. The CSAm source area for each sub-region is situated approximately 20° latitudes south of CSAl. Furthermore, the coverage range of CSAm is more limited and extends farther westward compared to CSAl (Fig. 8).

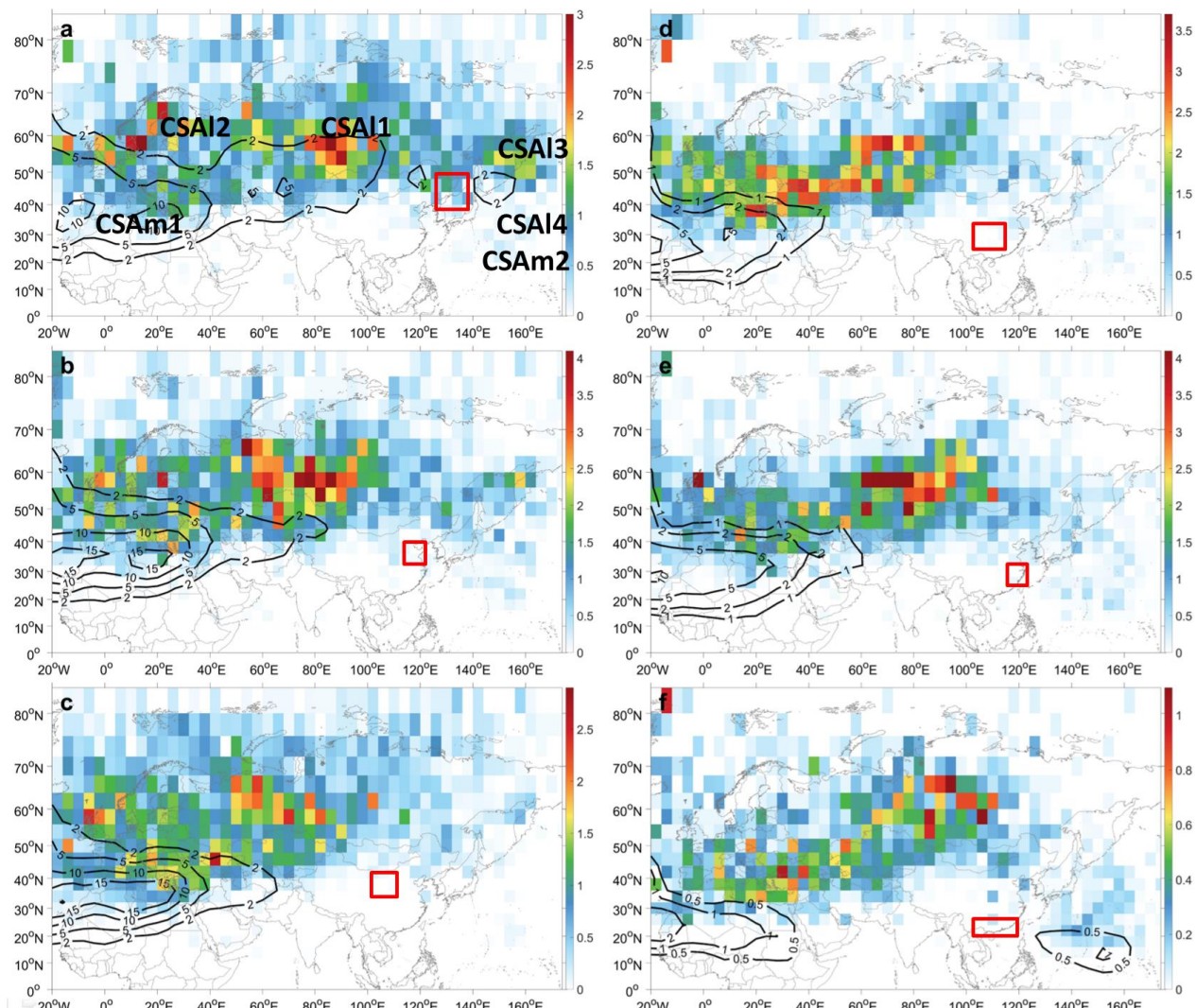

**Figure 8: Distributions of CSAII for ISI intruding the lower troposphere (color shaded) and middle troposphere (black lines) of a) NE, b) NC, c) LP, d) CC, e) EC and f) SC from May to August in 2019. Each sub-region's position is indicated with a red square.**

The most significant impact on the middle troposphere over each sub-region is attributed to CSAm1 (20 ° W-100 ° E, 15 ° -50 ° N), covering an extensive area from Europe to Central Asia (Fig. 8). CSAm1 is considerably distant from the six sub-regions, and the transport distance of stratospheric ozone from CSAm1 exceeds 10,000 kilometers. Table S2 provides the numerical values of CSAII for each CSA, detailing their impact on the middle and lower troposphere in the six sub-regions. CSAm1 exerts the greatest influence on the middle troposphere of LP, followed by NC, NE, EC, and CC, with minimal impact on SC. Another CSA affecting the middle troposphere is CSAm2 (130 ° -160 ° E, 10 ° -22 ° N), which has a marginal impact on the southeast coast of China and negligible influence on the northern and western sub-regions NE, NC and LP.

The CSAs that exert the most significant impact on the lower troposphere (CSAl) are concentrated in the longitude range of 20° W to 140° E and the latitude range of 40° to 70° N, displaying a zonal distribution that spans a vast area from Europe to East Asia (Fig. 8). Akritidis et al. (2021) conducted an analysis of a 16-year global climatology of tropopause folding, concluding that there are no significant deep fold regions over the Eurasian continent during the spring and summer seasons. This absence might be attributed to the challenges of effectively capturing the high-frequency variations of tropopause folding and SIs with low-resolution data in their climate averages. Within this defined region, three prominent CSAs emerge: CSAl1 (50° -100° E), CSAl2 (15° -45° E), and CSAl3 (135° -165° E). These three CSAs exhibit almost equal spacing, with an average interval of 70° longitudes. This zonal spacing aligns with the wavelength of synoptic-scale Rossby waves (Holton and Hakim, 2013), indicating that, besides the Brewer-Dobson circulation, the periodic oscillations of weather systems play a crucial role in driving regional-scale SIs. Among these three CSAs, CSAl1, located in Western Siberia, has the most substantial impact on the lower troposphere, with the most significant effect on NC, followed by the EC, LP, CC, and NE.

Conversely, SC experiences the least impact. CSAl2, situated in northern Europe, exerts considerable influence on CC, the LP, EC, and NC, while its impact on NE is negligible, and its effect on SC is minimal due to its comparative geographical position. Although CSAl3 has the least impact among the three CSAs, it has a certain influence on NE and NC, while its impact on the other four sub-regions of CEC is minimal due to the northeast location of CSAl3. Additionally, CSAl4, located in the Western Pacific between 135° -165° E and 15° -40° N, exerts a notable impact on coastal regions such as NC, EC, and SC. The long-range ozone descent from CSAs often manifest as horizontally wavelike transport paths, as previously discussed in our earlier research (Meng et al., 2022b). The dynamics underlying these long-range intrusions and their influence on tropospheric ozone necessitate a meteorological explanation.

Figures S1-S8 illustrate the spatial distribution of CSAs of ISI from May to August on a monthly basis. During this period, there is a discernible trend in the influence of CSAl migrating from northwest to southeast (Figs. S1-S4). This observed trend suggests a gradual weakening of the impact of CSAl1 and CSAl2 in the west side, while the influence of CSAl4 in the southeast strengthens. This qualitative change aligns with the jet-stream movement over the CEC during those periods. The temporal variation in CSAl4 implies that the influence may stem from the low-latitude easterly circulation system, with typhoons being a primary representative system. Although typhoons are infrequent weather phenomena and CSAl4 exerts limited effects, the intensity of such systems, rather than their occurrence frequency, is a crucial determinant of their impact. In our preceding study, it was discovered that the northward progression of Typhoon Ampil in 2018 resulted in a one-third surge in near-surface ozone within the North China Plain (Meng et al., 2022a). CSAm do not exhibit the same level of fluctuations as those witnessed in CSAl.

### 3.4.2 CSAs for DSI

The primary source areas, denoted as 2CSAs, of DSI impart a relatively modest influence on each sub-region of the CEC (Fig. 9). These sources exhibit a concentrated spatial coverage with a transport distance of DSI ozone confined within 60 longitudes. Similar to Figure 8, Figure 9 employs solid lines and color shading with the same meanings. Similarly, the crucial sources of DSI affecting the middle and lower troposphere are spatially distinct. 2CSAm, influencing the middle troposphere over each sub-region of CEC, is situated further south and west compared to 2CSAl, which affects the lower troposphere. Notably, 2CSAm1, positioned at 50°-90° E, 25°-45° N, originates over the Tibetan Plateau, influenced by the South Asian High modulated by the heat source over the plateau (Wang et al., 2021). This source exerts the most substantial impact on LP, followed by the sub-regions NC, NE, EC, and CC. In contrast, 2CSAm2 (95°-115° E, 20°-35° N) has the most pronounced influence on EC, followed by NC and CC, while SC experiences minimal impact.

Conversely, the key source areas 2CSAls of DSI exhibit little impact on the lower troposphere in six sub-regions of CEC. The monthly variation of 2CSAls is not statistically significant, and these sources are relatively concentrated in Mongolia and central Russia (Figs. S5-S8), indicating that the evolution of 2CSAls is constrained by the intense subsidence motion of regional typical weather systems.

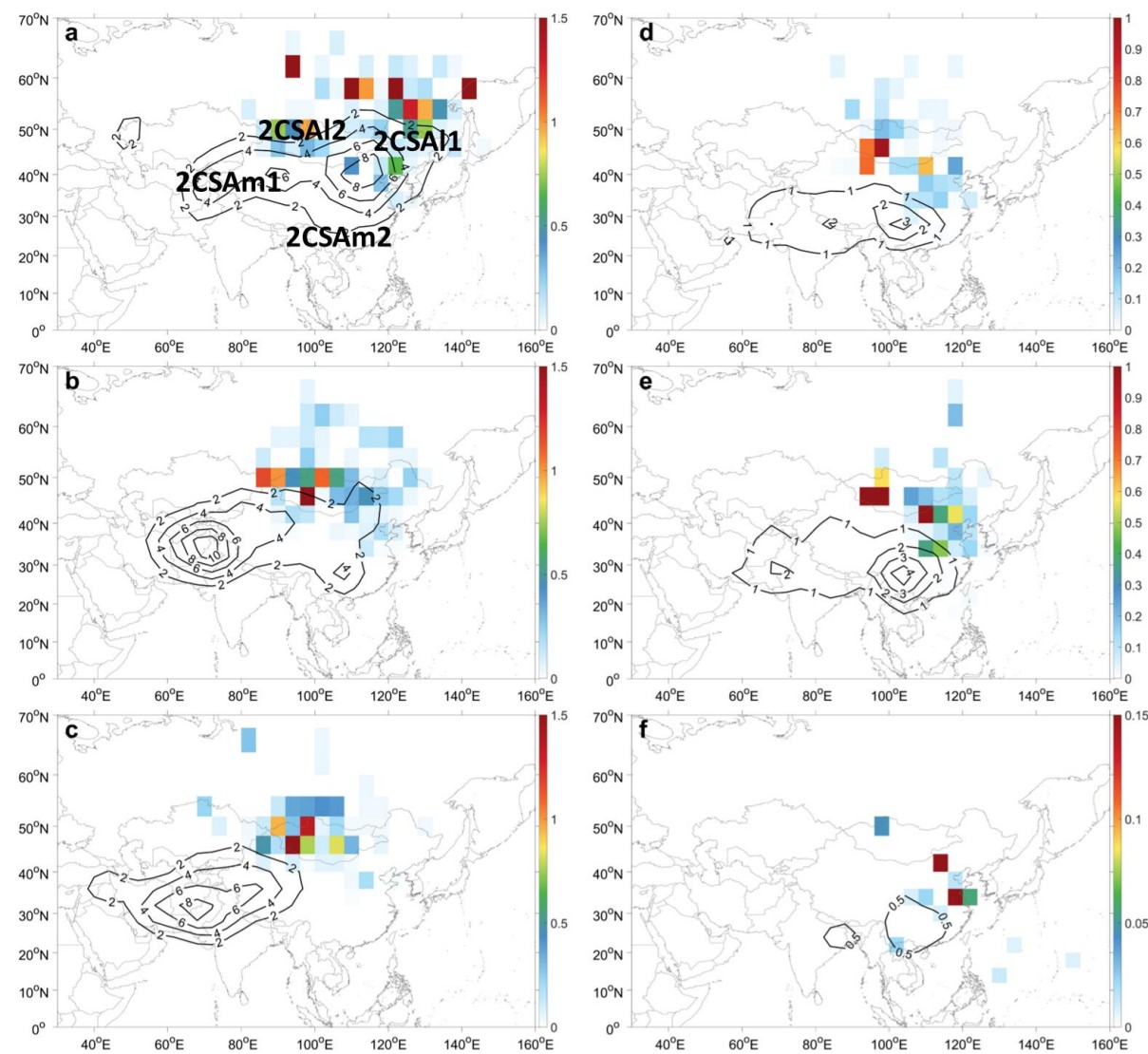

**Figure 9: Same as Fig. 8 but for DSI.**

### 3.5 Associated atmospheric circulations as driving factors for CSAs

Figure 10 illustrates the atmospheric circulation during May, June, July, and August in 2019 in the upper troposphere-lower stratosphere (UTLS). Comparing with the spatial distribution presented in Figure 8, it is evident that the CSAls and CSAms of ISI are intricately linked with the pattern of atmospheric fluctuations and are located within the subtropical jet in the UTLS, respectively. To exemplify, we focus on three CSAls corresponding to typical large-scale low-pressure systems impacting the lower tropospheric ozone: CSAl1 corresponds to Western Siberia, a region known for the formation of large-

scale weather systems, periodically generating robust and profound low-pressure systems (e.g., Figure 8b and 10b). Trickl et al. (2023) posit that Siberia is also the primary source area for SI affecting Central Europe.

CSAl3, located in Northeast China and eastern Russia, is associated with the frequent activity of the Northeast Cold Vortex. The stratospheric ozone emanating from CSAl4 over the western Pacific is propelled by the easterly circulation and transported from east to west across the western Pacific, reaching the coastal regions of eastern China.

The most significant source impacting the middle troposphere, CSAm1, originates from subtropical westerly jets over Europe and North Africa (south of 40°N). Conversely, stratospheric sources CSAls (north of 40°N) impacting the lower troposphere are more closely linked to large-scale fluctuations at mid to high latitudes. The significant spatial disparities in SI sources affecting mid and lower troposphere can be attributed to atmospheric circulation patterns in the Northern Hemisphere (Fig. 10). The meridional extent is greater at mid to high latitudes compared to lower latitudes, facilitating

southward transport of stratospheric air affecting the lower troposphere, while stratospheric air affecting the middle

troposphere is transported eastward in zonal westerlies with less meridional extent. These latitudinal variations in circulation patterns are pivotal in shaping distinct SI sources and transport pathways over CEC.

However, a comparative analysis between Figure 9 and Figure 10 reveals that the source areas of DSI do not exhibit a clear correlation with the positions of large-scale low pressure. This discrepancy arises from the intermittent and rarely independent nature of DSI events, in contrast to the continuous occurrence observed in ISI. Consequently, the average atmospheric circulation fails to distinctly reflect the correspondence between the source areas of DSI and weather systems. To delve deeper, we select a representative DSI event during May 19-20 that covered all the sub-regions except the sub-region SC. This specific case serves to meticulously unveil the driving role of the three-dimensional atmospheric structure in the DSI process.

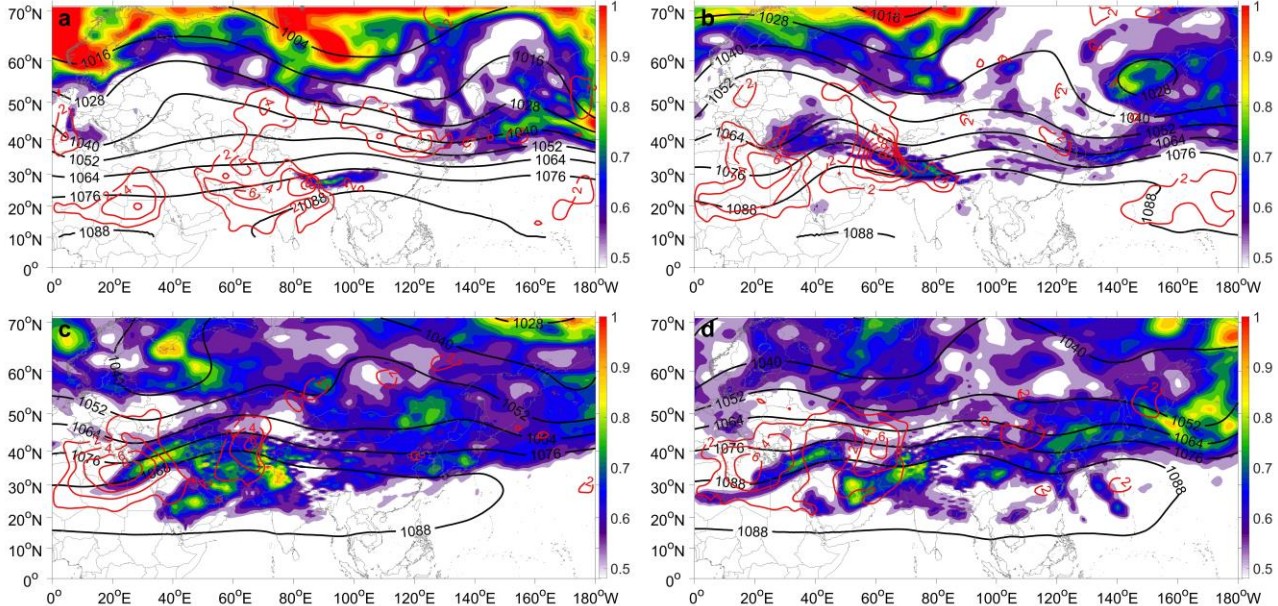

Figure 10: 400 hPa PV (PVU, color contours), geopotential height (gpdm, black lines), and vertical velocity (>0 hPa s-1, red lines) averaged at heights of 150–400 hPa in May (a), June (b), July (c) and August (d) in 2019.

Figure 11 illustrates a substantial meridional low-pressure system positioned within 2CSAl (Fig. 9), commonly recognized as the Northeast Cold Vortex (NECV). Here, we utilize this typical SI event to visually illustrate the transport process of ozone from the stratosphere into the free troposphere and subsequently into the atmospheric boundary layer driven by the NECV. This vortex stands as a prominent atmospheric system influencing the CEC, characterized by its prolonged presence as a deep, cyclonic low-pressure system of considerable intensity, primarily occurring in the spring and summer seasons. Concurrently, a robust subtropical westerly jet resides on the southern flank of the NECV (30°-40° N). On May 19, the SI ozone was predominantly situated north of 35° N, with the majority of stratospheric air particles concentrated within the middle troposphere (Figure 11a). By May 20, the stratospheric air particles exhibit southeastward movement, following the trajectory of the NECV (Figure 11b). Subsequently, a substantial portion of stratospheric ozone has permeated the lower troposphere, with some particles detected at altitudes below 500 m. On the 20th, an observed increase in near-surface ozone and a reduction in CO persisted for an average of 8 hours across vast areas in central CEC, which is also reflected in the distribution of stratospheric air in the boundary layer and the spatial correspondence of the SI vertical structure (Figures 10b and 10d). The vertical depiction of atmospheric circulation and ozone concentrations in Figures 11c and 11d underscores the deep tropopause folding near the NECV, extending downward to 500 hPa. A robust downdraft is evident below the folding zone, facilitating the descent of high-level stratospheric ozone towards the boundary layer (Fig. 11d). This aligns with the simulated trajectory in this study (indicated by the symbol '+' in Figure 11b). The impact of this DSI event primarily affected

the LP, NC, CC, and EC regions, resulting in SI ozone concentrations of 11.4, 6.1, 6.1, and 11 ppbv, respectively. NE also experienced the effects of the DSI event, observing SI ozone concentrations of 4.7 ppbv. However, the influence on SC remained negligible, with SI ozone levels below 0.4 ppbv. A prior study by Chen et al. (2014) highlighted the propensity of the NECV to induce stratosphere-troposphere exchange over the NE region. Our study contributes to a new understanding by revealing that the NECV can provide an exceptional pathway for ozone transport across the tropopause and a deep intrusion, leading to significant DSI impacts of daily scales in CEC.

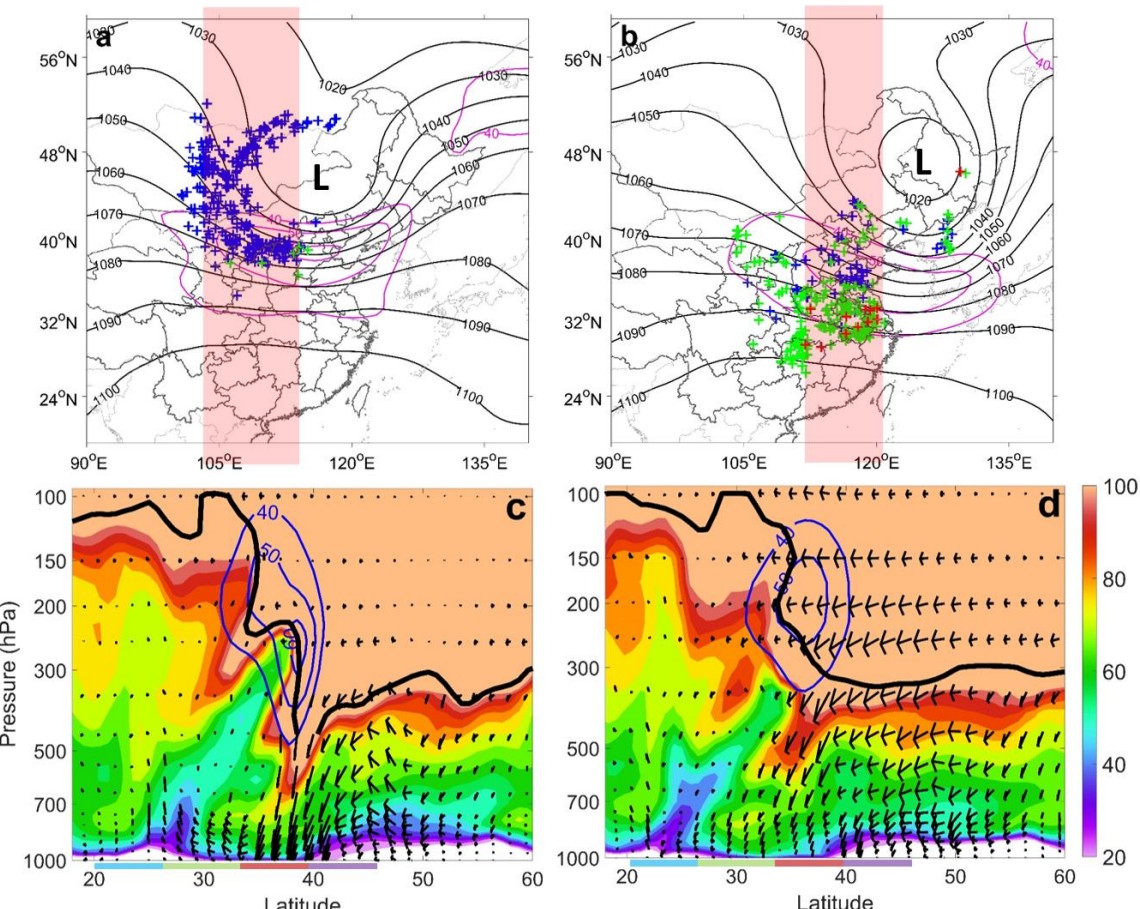

**Figure 11: 300-hPa geopotential height (gpdm, black lines) and U-component of horizontal wind at 200 hPa (> 40 m s− 1, magenta lines) at (a) 20:00 on May 19 and (b) 20:00 on May 20, 2019. The plus signs in Figures a and b represent the position of stratospheric air particles in the middle troposphere (blue), the lower troposphere (green) and below 500 m (red), respectively. Vertical cross section of wind speed (> 40 m s− 1, blue lines) and O3 concentrations (ppb, color contours) averaged along the red square in Figures a and b on (c) May 19 and (d) May 20, 2019. The thick solid black line in Figures c and d indicates the PV of 1.5 PVU. The arrows in Figures c and d are the V–W (meridional) wind fields. The bottom lines of purple, red, green and blue in Figures c and d indicates the sub- regions NE, NC and LP, CC and EC, and SC, respectively.**

## 4 Conclusions and Discussion

The continual investigation of the stratospheric contribution to tropospheric and near-surface ozone remains a critical research topic. The CEC region exhibits a consistent pattern of maximum ozone concentrations during the spring and summer seasons, as evidenced by observations of near-surface and tropospheric ozone. In this study, we delved into the correlation between the spring-summer ozone and the occurrence of SI in six sub-regions of CEC. Particular attention has been given to evaluating the contribution of SI to tropospheric and near-surface ozone and determining the source areas of SI critical to the CEC. This investigation represents the first attempt to define a systematic analysis of SI in the CEC and the differences in the impact of SI on different sub-regions. We also conducted detailed evaluations of the proposed SI assessment method in three aspects: comparing the spatial structure of simulated SI ozone with tropospheric ozone from OMI remote sensing data, examining the correlation between simulated SI ozone and stratospheric ozone tracers, and

comparing the SI events identified by the simulation with those identified from ground-based records.

Whether in DSI or ISI process, the tropospheric ozone in LP, NC, and NE is most significantly influenced by SI. In contrast, the impact of SI on the troposphere above the southern part is relatively less pronounced, with minimal influence on the troposphere of SC. This emphasizes that regions near the source areas (CSAs) of SI are more profoundly affected. Ozone transported indirectly to the troposphere registers twice the magnitude of direct transport, and ISI's impacts on tropospheric ozone over CEC are more efficient at monthly scales, while DSI's impacts are more pronounced at daily scales. The characteristics of tilted long-distance transport of stratospheric ozone are emphasized, which is determined by atmospheric baroclinicity. The frequencies and durations of DSI events in all sub-regions were notably lower than those of ISI, primarily occurring in May. LP was the sole region experiencing DSI events in July and August, while SC did not exhibit significant DSI events. In contrast, ISI events are more prevalent in the summer, associated with the development of well-established large-scale low-pressure systems in regions corresponding to summer CSAS.

Previous studies have fostered the widespread belief that SI is only significant at high-elevation sites (Attmannspacher and Hartmannsgruber, 1973; Ambrose et al., 2011; Stohl et al., 2000; Trickl et al., 2010, 2014). However, our research has shown that the most substantial contribution to near-surface ozone from ISI is observed in the eastern plains (NE, NC, EC, etc.), while contributions to LP and CC, located in the central and western high-altitude regions, are minimal. These differences in the contribution of SIs to ground-level ozone may be linked to regional variations in atmospheric boundary layer mixing and chemistry and their nonlinear complex interactions. The turbulent entrainment by the deep convective boundary layer in the eastern plains is likely more adept at capturing descending lower stratospheric air, playing a pivotal role in modulating near-surface ozone compared to the western parts of the country. This contributes significantly to the spring-summer ozone maximum in the eastern sub-regions. It is noteworthy that one cannot dismiss the possibility that near-surface ozone may also decrease during SI, especially during DSI events.

Understanding the stratospheric sources of tropospheric and near-surface ozone holds significant societal and climatic importance. In this study, we propose a traceability assessment method for SI, yielding high-resolution CSAs at a resolution 0.3° x 0.3°. Previous research, utilizing coarse grid data, has failed to reveal deep SI phenomena (Bartusek et al., 2023). The high-resolution CSAs obtained in this study enable us to scrutinize the layout of stratospheric sources influencing various sub-regions within the CEC region from a more refined and regional perspective. Whether involving DSI or ISI processes, the CSAs influencing the middle and lower troposphere originate from distinct sources. Specifically, the stratospheric air entering the middle troposphere and passing through a certain sub-region typically does not penetrate further into the boundary layer of that sub-region. Instead, it continues downstream with atmospheric circulation, a behavior determined by atmospheric baroclinicity. Since ozone has a relatively short lifetime in the troposphere (~ 3 weeks) and is not well mixed, stratospheric ozone can be transported from thousands of kilometers away through tilted pathways, influencing near-surface ozone. The distribution of CSAs influencing the lower troposphere for ISI, evenly distributed latitudinally within the westerlies (spaced 70 longitudes apart), is closely tied to the evolution of large-scale low-pressure systems, with Western Siberia being the most significant source. Siberia also stands out as a primary stratospheric source influencing Central Europe (Trickl et al., 2023). However, the most significant sources impacting the middle troposphere originate from subtropical westerly jets over Europe and North Africa.

In contrast, the CSAs for DSI cover a small range, with the CSAs influencing the middle troposphere associated with the South Asian High over the Tibetan Plateau, and the CSAs influencing the lower troposphere concentrated in Mongolia and central Russia. The impact of DSI arises from isolated and discontinuous weather systems, where the intensity of the intruding system plays a more pivotal role than its frequency of occurrence. Crucial drivers of DSI impacting the lower tropospheric ozone over CEC include weather systems such as the northeast cold vortex and typhoons, which cannot be overlooked.

It is worth noting that although the existing FLEXPART model does not account for tropospheric chemistry, due to the

limited free tropospheric mixing, Lagrangian approaches such as FLEXPART appear more promising than Eulerian models. The uncertainties inherent in using relatively coarse spatial resolution models and localized measurements may be the crucial constraints on stratosphere-to-troposphere transport studies. To achieve a comprehensive understanding of the seasonal and interannual variations of SIs and their critical source areas, as well as their influence on major invasion hotspots globally, further research utilizing high-resolution models and long-term extensive observations, including coordinated surface, remote sensing, and aircraft measurements, is imperative, especially in understanding the underlying physical mechanisms influencing variations in SI contribution to ground-level ozone in different regions. Furthermore, we are conducting SI ozonesonde experiments during the summer monsoon season (May-August) of 2024 to further validate our simulation and computation methods in future studies.

*Data availability.* All data used in this paper can be provided upon request from Kai Meng (macka@foxmail.com)

*Author contributions.* KM conceived the analysis and modeling study. KM designed the graphics and wrote the manuscript with help from TZ, MW and YJ conducted the observational data processing, YB. LC, MW, XH, and YL were involved in the scientific discussion. All authors commented on the paper.

*Competing interests.* The authors declare that they have no conflict of interest.

*Financial support.* This research was supported by the National Key Research and Development Program of China (Grant No. 2022YFC3701204) and the National Natural Science Foundation of China (Grant No. 42475195).

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
