# Peer review of "Tracing the origins of stratospheric ozone intrusions: direct vs. indirect pathways and their impacts on Central and Eastern China in spring-summer 2019"

_EGUsphere, 2024_

## Author Comment (AC1)

Dear Editors and Reviewers,

Thank you very much for your careful review on our manuscript egusphere-2024-930. We appreciate very much your constructive suggestions on improving our manuscript. We have accordingly made the careful and substantial revisions. The revised portions are marked up in the revised manuscript. Please find our point to point responses to the reviewers' comments as follows:

**Responses to the reviewer #1**

*General comments*

*[ This paper focuses on the impact of stratospheric intrusions (SI) on tropospheric ozone variations in China concerning the origins and transport pathways of these ozone-rich stratospheric airmasses. They rely on pure trajectory simulations using the FLEXPART model, through which a forward trajectory history of air parcels reaching the lower troposphere is obtained. Overall, the paper is well organized and written and fit the scope of the ACP journal. However, there are some issues that need to be addressed before it turned into a publication, especially in the context of the trajectory simulations and their validation and interpretation. More details of the trajectory settings, sensitivity analysis and validation should be provided in this paper. It would be better that the authors take more time to address these problems and the characteristics of the two SI types. So, I would like to suggest a major revision and hope to get a high-quality paper concerning the importance of SI in tropospheric ozone. ]*

**Response 1:** Many thanks for the encouraging comments and helpful suggestions on our manuscript. Following the reviewer's suggestions and comments, we have accordingly made careful revisions. Please find our point-to-point responses as follows:

*[ Major issues*

*1. The trajectory simulations rely heavily on the stratospheric tracer definition. In this paper, a PV value of 1.5 PVU was chosen to represent the dynamic tropopause and the stratospheric origin of airmasses. In my opinion, this value cannot sufficiently resolve the STE processes over such a broad region, though several researches have used a threshold of 1.5 or 1.8 PVU. For example, tropopause height is higher in warm months than in cold months, but the seasonal variations in tropopause height in South China derived from the 1.5-PVU dynamic tropopause are nearly constant throughout the year. In Fig. 7, the vertical distribution of stratospheric air parcels reaching the middle troposphere is southward placed than those reaching the PBL and ground surface. Such a distribution is opposite to the mechanism of stratospheric airmass transportation through tropopause folding where the ozone-rich airmasses are transported southward and downward (as shown in Fig. 10). In this perspective, the choice of 1.5 PVU seems to be not appropriate and may include airmasses in the upper troposphere (not the stratosphere) reaching the lower troposphere. I suggest that the authors should address the performances of different*

*PV thresholds to track stratospheric origins through trial-and-error analysis. Sensitivity tests are necessary to be performed in order to obtain reliable results over China. ]*

**Response 2:** Thank you for your detailed comments on our manuscript. We agree with the reviewer's opinion that tropopause height varies significantly with latitudes and seasons. Indeed, a fixed threshold of 1.5 PVU may not sufficiently resolve the STE processes across such a broad region in China. Therefore, based on the tropopause distribution characteristics outlined in Kunz et al. (2006) and Chen et al. (1995), we have introduced latitude-variable potential vorticity (PV) thresholds with 1.5 PVU north of 30°N and 3.0 PVU south of 30°N to define the dynamical tropopause and assesses the stratospheric origin of air masses over the study region in China in the revised manuscript. This updated approach confines ozone exclusively to stratospheric origins, crossing the tropopause into the troposphere. Moreover, this study investigates the stratospheric ozone intrusions during the season of East Asian summer monsoon from May to August 2019, we have not considered the seasonal change in tropopause height during the study period (May-August) with season-variable potential vorticity (PV) thresholds. We have modified Section 2 "Data and Methods" in the revised manuscript (Lines 143-176) to introduce this updated approach with latitude-variable potential vorticity (PV) thresholds as follows:

**2.2 Model configuration** We conducted a daily rolling simulation with 15-day forward trajectories of over 500,000 air particles released from the stratosphere over the Eurasian region (-20°W–180°E) between May 1 and August 31, 2019. In our FLEXPART simulation, we employed domain-filling technology (Chen et al., 2012; Drumond et al., 2010) to simulate the stratospheric ozone tracer, ensuring that only air masses released from the stratosphere were considered. During the initial stages of simulation, particles in the troposphere were filtered out, while stratospheric particles were assigned mass according to the formula: $M_{O_3} = M_{air} \cdot P \cdot C \cdot 48/29$, where $M_{air}$ denotes the air mass of the particle, P is the potential vorticity, C=60×10$^{-9}$ pvu$^{-1}$ is the ozone/PV scaling factor (Stohl et al., 2000, 2005), and the factor 48/29 is applied to convert volume to mass mixing ratio. Throughout the integration, particles were advected from the stratosphere into the troposphere driven by GFS wind fields. In mid-latitudes, the potential vorticity (PV) is commonly regarded as an indicator of the dynamical tropopause (Bellevue et al., 2007). PV values ranging from 1.0 or 1.6 PVU (Stohl et al., 2000) to 3.5 PVU (Hoerling et al., 1991) are typically used (Akritidis et al., 2016, 2018, 2021; Hoskins and Berrisford, 1988; Skerlak et al., 2014; Sprenger et al., 2003; Stohl et al., 2003). Gerasopoulos et

al. (2006), for instance, used a PV greater than 1.5 PVU in backward trajectories to track air masses of stratospheric origin. In this study, we defined the dynamic tropopause as a PV of 1.5 PVU and stratospheric air as a PV greater than 1.5 PVU. The model output included three-dimensional ozone concentrations and position information of particles at 3-hour intervals, a horizontal resolution of 0.3° × 0.3°, and 17 vertical layers spaced 500 m apart.

**2.3 CSA identification and CSA impact index**

We employed the three-dimensional trajectory information obtained from the FLEXPART simulation to identify particles that reached the middle and lower troposphere within each sub-region of CEC, either directly or indirectly from the stratosphere. Unique IDs were then assigned to these trajectories/particles, enabling seamless tracking during subsequent analyses. By assimilating the meteorological field with the trajectory information and applying the distance weighting method, we determined the spatial location of each trajectory as it crossed the tropopause. The PV values, interpolated from meteorological reanalysis data, were used to determine the dynamical tropopause. Specifically, we adopted latitude-dependent PV thresholds with 1.5 PVU north of 30°N and 3.0 PVU south of 30°N to define the dynamical tropopause and assesses the stratospheric origin of air masses over the study regions in China. This approach constrained ozone to originate exclusively from the stratosphere, crossing the tropopause into the troposphere. This methodology enabled us to identify the critical source areas (CSAs) of SI, which are the zones where stratospheric air masses penetrate into the troposphere by crossing the tropopause. They are characterized by a high proportion of gridded particles/trajectories information within these spatial locations.

Following the reviewer's suggestion, sensitivity tests with different PV thresholds to track stratospheric origins have been conducted in order to obtain reliable results over China, newly proposing the latitude-dependent PV thresholds for this study (New Figure 7). Compared to the original 1.5-PVU threshold, the latitude-dependent PV thresholds present reasonable performance with slight variations in the CSAII index over the sub-region of South China (Fig. 7f), influenced by the SI source above the northwest Pacific. The latitude-variable PV thresholds (New Figure 7) show less deviation from the 1.5-PVU threshold over other sub-regions in CEC, confirming the robustness of our simulations and CSAII index algorithm.

[Figure]

New Figure 7: Distributions of CSAII for ISI intruding the lower troposphere (color shaded) and middle troposphere (black lines) of over the regions a) NE, b) NC, c) LP, d) CC, e) EC and f) SC from May to August in 2019. Each sub-region's position has been indicated with the red square.

Regarding Figure 7, we have further clarified the definitions of direct and indirect SI following the reviewer's suggestion (see Response 6). In the revised manuscript (Lines 414-420), we propose that stratospheric ozone intrusions originating from subtropical westerly jets over Eurasia (south of 40 °N) affect the middle troposphere over CEC, whereas those from stratospheric sources (north of 40 °N) impacting the lower troposphere are closely linked to large-scale atmospheric circulation fluctuations over mid to high latitudes. These spatial disparities in SI sources affecting middle and lower troposphere could be attributed to atmospheric circulation patterns in the Northern Hemisphere (Figure 9). The meridional extent is greater at mid to high latitudes compared to lower latitudes, facilitating southward transport of stratospheric air affecting the lower troposphere, whereas stratospheric air affecting the middle troposphere is transported

eastward along zonal westerlies with less meridional extent. We argue that these latitudinal variations in atmospheric circulation are pivotal in shaping distinct SI sources and transport pathways over central and eastern China.

**References:**

Chen P .Isentropic cross-tropopause mass exchange in the extratropics[J].Journal of Geophysical Research Atmospheres, 1995, 1001(D8).DOI:10.1029/95JD01264.

Kunz A , Konopka P ,R. Müller,et al.Dynamical tropopause based on isentropic potential vorticity gradients[J].Journal of Geophysical Research Atmospheres, 2011, 116(D1):-.DOI:10.1029/2010JD014343.

*[ 2. The validation of the filtered stratospheric trajectories and the associated vertical profiles of ozone are essentially important to infer the key features of those SI events. However, a detailed validation is lacking in this manuscript, which may greatly impair the accuracy and robustness of the results. Given the rareness of SI reaching the lower troposphere (Fig. 5a), it is warranted to examine the responses of surface ozone and other pollutants to the SI inferred from the forward trajectory simulations, providing direct observational evidence. ]*

**Response 3:** We appreciate the reviewer's suggestion. The validation of the stratospheric trajectories and the associated vertical profiles of ozone are essentially important to infer the key features of SI events. Due to the difficulties in the tropospheric observations of stratospheric ozone tracer, assessing stratospheric intrusions and their vertical structure poses inherent challenges. Following the reviewer's suggestion, we have validated the reliability of our FLEXPART simulations on stratospheric trajectories by comparing vertical structures of SI ozone and TOST ozone concentrations averaged monthly within the mid and lower troposphere. TOST ozone data are derived from global ozonesonde observations (Liu et al., 2013a). The comparisons between TOST ozone concentrations and SI ozone in sub-regions over CEC have been added in Table S1 in the revised supplement. The vertical variations of TOST and SI ozone exhibit similar temporal and regional patterns over CEC in the troposphere, and the simulated SI ozone concentrations in the troposphere over these sub-regions are notably lower than the TOST ozone concentrations, highlighting reasonable relations between stratospheric and total ozone (SI + tropospheric production) in the troposphere and underscoring the credibility of our simulation methodology. We have included the validation details in Section 3.2 (Lines 258-267).

Table S1 TOST and SI ozone concentrations (ppbv) in mid and lower troposphere over six sub-regions from May to August 2019

| | | NE | | NC | | LP | | CC | | EC | | SC | |
|---|---|---|---|---|---|---|---|---|---|---|---|---|---|
| | Month | TOST | SI | TOST | SI | TOST | SI | TOST | SI | TOST | SI | TOST | SI |
| **Middle troposphere (3-7 km)** | May | 68.5 | 22.2 | 71.1 | 23.3 | 70.6 | 31.8 | 69.5 | 19.3 | 69.2 | 19.9 | 64.4 | 5.7 |
| | June | 71.8 | 29 | 75.6 | 30.6 | 77.7 | 37.3 | 68.9 | 14.3 | 63.7 | 18.2 | 53.2 | 6.6 |
| | July | 68 | 32.4 | 74.2 | 37.7 | 73.9 | 44 | 55.4 | 17 | 49.8 | 18.8 | 48.2 | 4.8 |
| | August | 63.3 | 28.4 | 68.9 | 32.6 | 67.4 | 35.4 | 58.7 | 20.3 | 55.5 | 19.5 | 51.5 | 14.2 |
| **Lower troposphere (<=2 km)** | May | 57.2 | 7.3 | 63.7 | 9.7 | 64.8 | 16.2 | 61 | 10.4 | 59.2 | 9.5 | 54.2 | 3.3 |
| | June | 51.7 | 8 | 62.1 | 10.9 | 65.9 | 13.8 | 46.9 | 8.4 | 46.1 | 8.2 | 43.9 | 1.6 |
| | July | 42.1 | 5.5 | 45 | 8.7 | 36.8 | 18.1 | 36.2 | 8.6 | 40.4 | 4.2 | 39.7 | 0.7 |
| | August | 41.9 | 4.1 | 31.7 | 7.9 | 52.1 | 16.5 | 47.8 | 8.5 | 38.5 | 5.5 | 41 | 3.6 |

Furthermore, we are conducting SI ozonesonde experiments during the summer monsoon season (May-August) of 2024 to further validate our simulation and computation methods in future studies.

**References:**

Liu, G., Liu, J., Tarasick, D. W., Fioletov, V. E., Jin, J. J., Moeini, O., Liu, X., Sioris, C. E., and Osman, M.: A global tropospheric ozone climatology from trajectory-mapped ozone soundings, Atmos. Chem. Phys., 13, 10659-10675, 10.5194/acp-13-10659-2013, 2013a.

*[ 3. The authors claimed that they provided a "long-term" result of SI reaching middle-to-low troposphere in China. Only intrusions from May to August in 2019 are provided in this paper, while actually SI peak in later winter and early spring. The expression of "long-term" is not appropriate and may be biased because only several months in 2019 were included. A complete examination throughout a year, at least in 2019, are essentially important to reveal the seasonality of SI. ]*

**Response 4:** Thanks for the reviewer's comments. Only intrusions from May to August in 2019 are provided in this paper, while actually SI peak maybe in later winter and early spring. The objective of this study is to analyze maximum tropospheric ozone during late spring and summer 2019, particularly in relation to stratospheric intrusions. In response to the reviewer's suggestion, we have modified the title to "Tracing the origins of stratospheric ozone intrusions: direct vs. indirect pathways and their impacts on Central and Eastern China in spring-summer 2019".

*[ 4. Similar to Point 3, the authors only considered the Eurasia region in their domain-filling settings. As known, stratospheric air can travel thousands of kilometers during their downward descending. Thus, SI that initiate outside the Eurasia are likely to be missed and hence the stratospheric impact is underestimated to some extent. Please address this point and assess the sensitivity of the domain setting. ]*

**Response 5:** As noted by the reviewer, the chosen spatial extent is crucial for studying stratospheric intrusions. Our study investigates the relationship between tropospheric ozone over Central and Eastern China and stratospheric intrusions during late spring and summer of 2019. In a recent study, we conducted a 20-year statistical analysis of stratospheric intrusion source regions affecting the lower tropospheric ozone over China, with the sensitivity analyses on transport periods ranging from 1 to 30 days. As illustrated in the accompanying Figure R1, covering the entire Northern Hemisphere, stratospheric sources influencing lower-tropospheric ozone in China are primarily concentrated over the Eurasian continent, confirming the choice of spatial range (-20°W–180°E) in our domain-filling settings as reasonable and reliable. Therefore, there should be no missing SI that initiate outside of Eurasia for estimating the stratospheric impact.

[Figure]

**Figure R1** stratospheric intrusion source regions affecting the lower troposphere over China, with sensitivity analyses for transport periods ranging from 1 to 30 days.

*[ 5. The direct and indirect SI are divided according to the descending time. It is not clear how the descending time is determined. Is it the time of stratospheric air reaching the middle troposphere or the lower troposphere? From Fig. 3, I guess that the authors calculated the time when stratospheric air reaches the middle troposphere, since there are no direct intrusion cases reaching the boundary layer. Often, the direct intrusions refer to those descending rapidly and vigorously into the lower troposphere even the ground level, inducing strong perturbations to chemical species such as ozone, since the dilution and mixing of stratospheric air is reduced in shorter period. However, in this paper, the direct intrusions are weaker than the indirect ones, which is opposite to previous studies, e.g., the Fig. 4 of Cristofanelli et al. (2006). In this angle, the authors should rethink the way to separate direct and indirect SI in their trajectory samples. ]*

**Response 6:** We appreciate the reviewer's insightful suggestions. We have accordingly refined our definitions of direct and indirect stratospheric intrusions (Lines 214-227). In previous studies, stratospheric air was proposed to enter the lower troposphere either directly through rapid vertical movement (lasting less than 2 days) or indirectly from distant locations through a phased tilted mode (resulting in longer

tropospheric residence times, e.g., > 10 days) (Eisele et al., 1999; Meng et al., 2022b; Zanis et al., 1999). In this study, we have expanded on these concepts to redefine direct stratospheric intrusions, where stratospheric air from distinct sources reaches the mid or lower troposphere over a specific region (e.g. six sub-regions in this study, Fig. 1e) within approximately two days, and indirect stratospheric intrusions, where stratospheric air from distinct sources reaches these sub-regions' mid or lower troposphere within approximately ten days. It is noteworthy that either the mid or lower tropospheric layers serve as endpoints for stratospheric air transport in our definitions, with transit times set at two or ten days for these sub-regions, respectively. Furthermore, our definitions emphasize the concept about sources of stratospheric intrusion.

Additionally, we have carefully reviewed pertinent details as documented in Cristofanelli et al. (2006), which are incongruent with the conclusions of our study. We attribute this incongruence to differing definitions of direct and indirect intrusions, as well as divergent methodologies for identifying intrusion types. In our approach, we consistently emphasize the spatial concept of critical source areas (CSAs), where stratospheric air traverses the tropopause into the middle or lower troposphere. Previous studies have not considered the sources of stratospheric ozone, relying solely on near-surface environmental-meteorological changes and back-trajectory simulations to screen direct and indirect stratospheric intrusions. As shown in Figures 7 and 8 of this study, sources of indirect intrusions have a greater impact on tropospheric ozone compared to direct intrusions (noting that indirect intrusions cover a larger spatial extent and exhibit a greater impact intensity). This could be attributed to significant transport of stratospheric particles affecting the lower troposphere from sources of indirect intrusions, dictated by spatial characteristics of atmospheric circulation. For instance, one prominent source area identified in our study is over Siberia, which is also recognized as a primary source region for SI impacting Central Europe, as hypothesized by Trickl et al. (2023).

**References**

Trickl, T., Couret, C., Ries, L., and Vogelmann, H.: Zugspitze ozone 1970–2020: the role of stratosphere–troposphere transport, Atmos. Chem. Phys., 23, 8403-8427, 10.5194/acp-23-8403-2023, 2023.

*[ 6. The TOST ozone reanalysis profile data, with a horizontal resolution 5 degrees, rely heavily on the available ozonesonde observations. These data are not sufficient and appropriate for regional-scale analysis of vertical ozone distributions, especially over those ozonesonde-lacking regions. ]*

**Response 7:** The resolution of TOST data is limited to a horizontal resolution 5 degrees (Liu et al., 2013b). In this study, we employed this data primarily for background climatic concentrations in the lower stratosphere when calculating the CSAII index.

**References**

Liu, J., Tarasick, D. W., Fioletov, V. E., McLinden, C., Zhao, T., Gong, S., Sioris, C., Jin, J. J., Liu, G., and Moeini, O.: A global ozone climatology from ozone soundings via trajectory mapping: a stratospheric perspective, Atmos. Chem. Phys., 13, 11441-11464, 10.5194/acp-13-11441-2013, 2013b.

*[ 7. Line268-270: In Fig. 3, there is no contribution of SI to ozone below 2 km, and hence the calculation of near-surface ozone associated with direct SI (green boxes in Fig. 6) is totally mismatched. Moreover, even for those indirect intrusions (red boxes in Fig. 6), highest value was not seen in LP. I suggest that the authors should pay more attention to the interpretation of the results. ]*

**Response 8:** We greatly appreciate the reviewer's suggestions. We are sorry for not indicating the altitude above sea level (a.s.l.) in Figure 3, which has been added in the revised caption of Figure 3.

The interruption of the red line in Figure 3 below 1 km does not imply that stratospheric intrusions do not contribute below 2 km (a.s.l.), but rather reflects the higher altitude of the LP region above 1 km (a.s.l.). The LP region is notably characterized by higher altitudes compared to other five sub-regions. Figure 3 presents vertical profiles of stratospheric contributions directly transported to the troposphere, while Figure 6 illustrates near-surface ozone observations extracted during daytime and nighttime for direct, indirect intrusions, and intrusion-free periods. It indicates that near-surface ozone levels in the LP during intrusion periods are generally lower compared to intrusion-free periods (see Figure 6), despite stratospheric ozone reaching the LP's mid and lower troposphere exceeding that of other sub-regions (see Figures 3 and 4). Trickl et al. (2023) examined observations at the Alpine station Zugspitze and also found that occurrences of SI do not consistently lead to an increase in ground-level ozone. Thus, the SI ozone entering the lower

troposphere does not necessarily equate to higher near-surface ozone levels, suggesting a nonlinear process. This phenomenon may be attributed to the limited efficiency of certain atmospheric boundary layers in capturing descending stratospheric air to enhance near-surface ozone.

*[ 8. There are too few samples of direct SI in Fig. 6 to perform a statistical analysis. ]*

**Response 9:** Thank you for your suggestion. 9 direct SI samples affecting LP, and 2-4 samples affecting the remaining five sub-regions over CEC, are used to perform a statistical analysis in Figure 6.

*[ 9. Fig.9-10. More ground-based observations, either chemical composition or meteorological data, are necessary to validate the occurrence of this SI. ]*

**Response 10:** Figure 10 and associated discussions aim to visually illustrate the transport process of ozone from the stratosphere into the free troposphere and subsequently into the atmospheric boundary layer during a typical SI event driven by the Northeast Cold Vortex. Figure 10, depicting circulation patterns and the spatial evolution of stratospheric particles along with their downward transport structure, adequately conveys our intended points. In response to the reviewer's comments, we have also added the following discussions in the revised manuscript (Lines 518-520):

With more ground-based observations, either chemical composition or meteorological data, future studies could be necessary to further validate the occurrence of SI in the troposphere.

---

## Author Comment (AC2)

Dear Editors and Reviewers,

Thank you very much for your careful review on our manuscript egusphere-2024-930. We appreciate very much your encouraging comments and constructive suggestions on improving our manuscript. We have accordingly made the careful and substantial revisions. The revised portions are marked up in the revised manuscript. Please find our point to point responses to the reviewers' comments as follows:

**Responses to the reviewer #2**

*[ This study investigates the impacts of stratospheric ozone intrusion on tropospheric ozone in central and eastern China, using air quality observations, global ozone (re)analysis data, and customized Lagrangian simulations. The results are interesting and the manuscript is well organized in general, however, there are still some problems need to be improved before a consideration for publication. ]*

**Response 1:** Many thanks for the encouraging comments and helpful suggestions on our manuscript. Following the reviewer's suggestions and comments, we have accordingly made careful revisions. Please find our point-to-point responses as follows:

*[ Abstract L16: SIs ]*

**Response 2:** Thank you for pointing out the printing errors. It has been corrected to SIs in the revised manuscript.

*[ Data: Why the former ERA-Interim data was used in this study, instead of the more advanced ERA5 or MERRA2? ]*

**Response 3:** We appreciate the reviewer's suggestions. In the revised manuscript (Lines 117-120), we have added accordingly the following discussions:

In this study, ERA data are solely used for large-scale atmospheric circulation analysis, as detailed in Section 3.5. Stratospheric intrusions are primarily controlled by large-scale circulation, and our analysis using ERA-Interim data at 0.75-degree resolution effectively addresses the scientific issue.

Nevertheless, future studies could utilize more advanced ERA5 or MERRA2 data to explore more scientific insights.

*[ Model configuration: Why the year 2019 was chosen to simulate? As I know, the SI's contribution to tropospheric ozone in China is significantly larger than normal years. Anyway, to discuss the SI's contribution with simulations only in one year will lead to large uncertainties due to the interannual differences. ]*

**Response 4:** Thanks for the reviewer's comments. In response to the comments, we have accordingly added the following discussions in the revised manuscript:

The year 2019 was chosen in this study as a sequel of our previous studies (Meng et.al. 2022a, b) on the influence of stratosphere-to-troposphere transport on summertime surface $O_3$ changes in 2019 and a typhoon-induced SI event in North China Plain in a series of research (Lines 102-105).

This study only in one year could lead to uncertainties due to the interannual differences. Further study with multi-years global simulations combining with diverse observation data could comprehensively understand the impacts of stratospheric intrusions on tropospheric ozone in China, encompassing multi-year climate, interannual and seasonal variations.

By the way, we have modified the title to "Tracing the origins of stratospheric ozone intrusions: direct vs. indirect pathways and their impacts on Central and Eastern China in spring-summer 2019".

**References**

Meng, K., Zhao, T., Xu, X., Hu, Y., Zhao, Y., Zhang, L., Pang, Y., Ma, X., Bai, Y., Zhao, Y., and Zhen, S.: Anomalous surface O3 changes in North China Plain during the northwestward movement of a landing typhoon, Sci. Total Environ., 820, 153196, https://doi.org/10.1016/j.scitotenv.2022.153196, 2022a.

Meng, K., Zhao, T., Xu, X., Zhang, Z., Bai, Y., Hu, Y., Zhao, Y., Zhang, X., and Xin, Y.: Influence of stratosphere-to-troposphere transport on summertime surface $O_3$ changes in North China Plain in 2019, Atmos. Res., 276, 106271, https://doi.org/10.1016/j.atmosres.2022.106271, 2022b.

*[ It is better to describe briefly about the domain-filling technology here. ]*

**Response 5:** Thanks for the reviewer's suggestions; we have supplemented a detailed description of the domain-filling technique in Section 2.2 in the revised manuscript (Lines 143-150) as below:

In our FLEXPART simulation, we employed domain-filling technology (Chen et al., 2012; Drumond et al., 2010) to simulate the stratospheric ozone tracer, ensuring that only air masses released from the stratosphere

were considered. During the initial stages of simulation, particles in the troposphere were filtered out, while stratospheric particles were assigned mass according to the formula: $M_{O_3} = M_{air} \cdot P \cdot C \cdot 48/29$, where $M_{air}$ denotes the air mass of the particle, P is the potential vorticity, C=60×10$^{-9}$ pvu$^{-1}$ is the ozone/PV scaling factor (Stohl et al., 2000, 2005), and the factor 48/29 is applied to convert volume to mass mixing ratio. Throughout the integration, particles were advected from the stratosphere into the troposphere driven by GFS wind fields.

**References**

Stohl, A., Forster, C., Frank, A., Seibert, P., and Wotawa, G.: Technical note: The Lagrangian particle dispersion model FLEXPART version 6.2, Atmospheric Chemistry & Physics, 5, 2461-2474, 2005.

Stohl, A., Spichtinger-Rakowsky, N., Bonasoni, P., Feldmann, H., Memmesheimer, M., Scheel, H. E., Trickl, T., Hübener, S., Ringer, W., and Mandl, M.: The influence of stratospheric intrusions on alpine ozone concentrations, Atmos. Environ., 34, 1323-1354, https://doi.org/10.1016/S1352-2310(99)00320-9, 2000.

*[ Figure 1: The colormap used in this figure should be updated to make the signals on a-b and k-m clearer. ]*

**Response 6:** We have replaced the colormap in Figure 1.

[Figure]

Figure 1: Distributions of the observed near-surface $O_3$ concentrations (ppbv) over CEC for (a-l) January to December during 2015-2019 with the sub-regions NE (NorthEast China); NC (North China); LP (Loess Plateau); CC (Central China); EC (East China); SC (Southern China).

*[ Figure 2: The vertical ozone gradients also represent the position of the tropopause. Seen from this figure, the 1.5 PVU may not a proper choice for all the selected regions. ]*

**Response 7:** Thank you for your detailed comments on our manuscript. We agree with the reviewer's opinion that tropopause height varies significantly with latitudes and seasons. Indeed, a fixed threshold of 1.5 PVU may not sufficiently resolve the STE processes across over such a broad region in China. Therefore, based on the tropopause distribution characteristics outlined in Kunz et al. (2006) and Chen et al. (1995), we have introduced latitude-variable potential vorticity (PV) thresholds with 1.5 PVU north of 30 °N and 3.0 PVU south of 30 °N to define the dynamical tropopause and assesses the stratospheric origin of air masses over the study region in China in the revised manuscript. This updated approach confines ozone exclusively to stratospheric origins, crossing the tropopause into the troposphere. We have modified Section 2 "Data and

Methods" in the revised manuscript (Lines 143-176) to introduce this updated approach with latitude-variable potential vorticity (PV) thresholds as follows:

**2.2 Model configuration**

We conducted a daily rolling simulation with 15-day forward trajectories of over 500,000 air particles released from the stratosphere over the Eurasian region (-20∘W–180∘E) between May 1 and August 31, 2019. In our FLEXPART simulation, we employed domain-filling technology (Chen et al., 2012; Drumond et al., 2010) to simulate the stratospheric ozone tracer, ensuring that only air masses released from the stratosphere were considered. During the initial stages of simulation, particles in the troposphere were filtered out, while stratospheric particles were assigned mass according to the formula: $M_{(O_3)} = M_{air} \cdot P \cdot C \cdot 48/29$, where $M_{air}$ denotes the air mass of the particle, $P$ is the potential vorticity, $C = 60 \times 10^{-9}$ pvu$^{-1}$ is the ozone/PV scaling factor (Stohl et al., 2000, 2005), and the factor 48/29 is applied to convert volume to mass mixing ratio. Throughout the integration, particles were advected from the stratosphere into the troposphere driven by GFS wind fields. In mid-latitudes, the potential vorticity (PV) is commonly regarded as an indicator of the dynamical tropopause (Bellevue et al., 2007). PV values ranging from 1.0 or 1.6 PVU (Stohl et al., 2000) to 3.5 PVU (Hoerling et al., 1991) are typically used (Akritidis et al., 2016, 2018, 2021; Hoskins and Berrisford, 1988; Skerlak et al., 2014; Sprenger et al., 2003; Stohl et al., 2003). Gerasopoulos et al. (2006), for instance, used a PV greater than 1.5 PVU in backward trajectories to track air masses of stratospheric origin. In this study, we defined the dynamic tropopause as a PV of 1.5 PVU and stratospheric air as a PV greater than 1.5 PVU. The model output included three-dimensional ozone concentrations and position information of particles at 3-hour intervals, a horizontal resolution of 0.3∘ × 0.3∘, and 17 vertical layers spaced 500 m apart.

**2.3 CSA identification and CSA impact index**

We employed the three-dimensional trajectory information obtained from the FLEXPART simulation to identify particles that reached the middle and lower troposphere within each sub-region of CEC, either directly or indirectly from the stratosphere. Unique IDs were then assigned to these trajectories/particles, enabling seamless tracking during subsequent analyses. By assimilating the meteorological field with the trajectory information and applying the distance weighting method, we determined the spatial location of each trajectory as it crossed the tropopause. The PV values, interpolated from meteorological reanalysis

data, were used to determine the dynamical tropopause. Specifically, we adopted latitude-dependent PV thresholds with 1.5 PVU north of 30 °N and 3.0 PVU south of 30 °N to define the dynamical tropopause and assesses the stratospheric origin of air masses over the study regions in China. This approach constrained ozone to originate exclusively from the stratosphere, crossing the tropopause into the troposphere. This methodology enabled us to identify the critical source areas (CSAs) of SI, which are the zones where stratospheric air masses penetrate into the troposphere by crossing the tropopause. They are characterized by a high proportion of gridded particles/trajectories information within these spatial locations.

Following the reviewer's suggestion, sensitivity tests of with different PV thresholds to track stratospheric origins have been conducted performed in order to obtain reliable results over China, newly proposing the latitude-dependent PV thresholds for this study (New Figure 7). Compared to the original 1.5-PVU threshold, the latitude-dependent PV thresholds present reasonable performance with slight variations in the CSAII index over the sub-region of South China (Fig. 7f), influenced by the SI source above the northwest Pacific. The latitude-variable PV thresholds (New Figure 7) show less deviation from the 1.5-PVU threshold over otherthe rest sub-regions in CEC, confirming the robustness of our simulations and CSAII index algorithm.

[Figure]

New Figure 7: Distributions of CSAII for ISI intruding the lower troposphere (color shaded) and middle troposphere (black lines) of over the regions a) NE, b) NC, c) LP, d) CC, e) EC and f) SC from May to August in 2019. Each sub-region's position has been indicated with the red square.

**References**:

Chen P .Isentropic cross-tropopause mass exchange in the extratropics[J].Journal of Geophysical Research Atmospheres, 1995, 1001(D8).DOI:10.1029/95JD01264.

Kunz A , Konopka P ,R. Müller,et al.Dynamical tropopause based on isentropic potential vorticity gradients[J].Journal of Geophysical Research Atmospheres, 2011, 116(D1):-.DOI:10.1029/2010JD014343.

*[ 3.2: Why the 2 and 10 days forward trajectories can indicate the direct and indirect transports? The reason should be well explained and the sensitivity of the results to the selected length should be tested. ]*

**Response 8:** In response to the reviewer's suggestion, we have clarified our definitions of direct and indirect stratospheric intrusions. Previous studies (Eisele et al.; 1999; Meng et al., 2022b; and Zanis et al., 1999) proposed that stratospheric air can be transported to the lower troposphere directly through rapid vertical movement (lasting less than 2 days in the troposphere) or indirectly from thousands of kilometers away through a phased tilted mode (resulting in longer tropospheric residence times, such as > 10 days). We have expanded upon these concepts to distinguish between direct and indirect stratospheric intrusions, where direct intrusions involve stratospheric air from specific sources reaching mid or lower troposphere regions within approximately two days, while indirect intrusions extend this period to around ten days. Notably, both mid and lower tropospheric regions are considered endpoints for stratospheric air transport in our definitions, with transit times set at two and ten days for these regions, respectively.

In the revised manuscript (Lines 214-221), we have added the above sentences.

**References**

Eisele, H., Scheel, H. E., Sladkovic, R., and Trickl, T.: High-Resolution Lidar Measurements of Stratosphere–Troposphere Exchange, Journal of the Atmospheric Sciences, 56, 319-330, 10.1175/1520-0469(1999)056<0319:HRLMOS>2.0.CO;2, 1999.

Meng, K., Zhao, T., Xu, X., Zhang, Z., Bai, Y., Hu, Y., Zhao, Y., Zhang, X., and Xin, Y.: Influence of stratosphere-to-troposphere transport on summertime surface O3 changes in North China Plain in 2019, Atmos. Res., 276, 106271, https://doi.org/10.1016/j.atmosres.2022.106271, 2022b.

Zanis, P., SCHUEPBACH, E., GÄGGELER, H. W., HÜBENER, S., and TOBLER, L.: Factors controlling beryllium-7 at Jungfraujoch in Switzerland, Tellus B, 51, 789-805, https://doi.org/10.1034/j.1600-0889.1999.t01-3-00004.x, 1999.

*[ Figure 3 and 4: How the contributions of stratospheric ozone directly/indirectly transported to the troposphere were estimated? As an important result of the paper, detailed description of the methos should be clearly described. ]*

**Response 9:** Thanks for the reviewer's suggestions. The contributions presented in this study originate from ozone concentration calculations at various altitudes within the troposphere obtained from our FLEXPART simulation. The calculation of concentration within each model grid is achieved by sampling tracer mass fractions of all particles within the grid cell and dividing by the grid cell volume:

$$C = 1 \Big/ V \cdot \sum_{i=1}^{N} (m_i f_i)$$

Here, V represents the volume of the grid cell, $m_i$ denotes the mass of particle i, N is the total number of particles, and $f_i$ represents the mass fraction contributed by particle i to the respective grid cell. The mass fraction $f_i$ is computed using a uniform kernel with grid distances $\Delta x$ and $\Delta y$ in the longitude-latitude output grid. We have incorporated the above methodology description into the revised manuscript (Lines 157-163).

Assessing stratospheric intrusions and their transport poses inherent challenges. In the revised manuscript, we validate the reliability of our FLEXPART simulations by comparing SI ozone and TOST ozone concentrations averaged monthly across the mid and lower troposphere layers. TOST ozone data are derived from global ozonesonde observations and Lagrangian models, as evaluated for reliability in Liu et al. (2013a). Concentrations of TOST and SI ozone in each sub-region are detailed in Table S1 in the revised supplement. SI ozone concentrations in each layer of each sub-region are notably lower than TOST values but exhibit similar temporal and regional trends. These comparative findings underscore the credibility of our simulation methodology.

*[ Figure 7: It is better to mark the location of each region. ]*

**Response 10:** As requested by the reviewer, the position of each sub-region has been marked in the new Figure 7 (please see response 7).

*[ P12 L305: Is the CSAm1 related to the southern Asia High or collocated with the subtropical jet? ]*

**Response 11:** We appreciate the reviewer's discussion. As depicted in Figure 9, CSAm1 corresponds to the subtropical jet over Europe and North Africa. Accordingly, we have updated the following description at lines 404-406: "it is evident that the CSAls and CSAms of ISI are intricately linked with the pattern of atmospheric fluctuations and are located within the subtropical jet in the UTLS."

*[ Figure 9: The colormap should be improved for this figure to make the information more visable. ]*

**Response 12:** Thanks for your careful review. We have improved Figure 9 to enhance the visibility of the information.

[Figure]

Figure 9: 400 hPa PV (PVU, color contours), geopotential height (gpdm, black lines), and vertical velocity (>0 hPa s-1, red lines) averaged at heights of 150–400 hPa in May (a), June (b), July (c) and August (d) in 2019.

---

## Author Response (AR2)

Dear Editors and Reviewers,

We apologize for not fully addressing your revision requests in our previous responses. We sincerely appreciate your meticulous review of our manuscript egusphere-2024-930. In response to your feedback, we have made significant and careful revisions, particularly concerning the validation of the methods used in this study. The revised portions are marked in the revised manuscript. Please find our responses to the reviewers' comments as follows:

**Responses to the reviewer #1**

*[This work investigates the stratosphere-to-troposphere transport (STT) processes based on pure trajectory simulations; however, reasonable validations of trajectory are totally absent. The representation of tropopause is a fundamental factor controlling where and how stratospheric air enters the troposphere, but it is poorly resolved in the manuscript that simply uses two fixed PV thresholds over very broad regions. Therefore, detailed validations of simulations against observations are essentially important for the reliability of results. Though the authors provide some comparisons with TOST ozone data, the coarse temporal and spatial resolution of TOST is not able to track the fine-scale features of STT. ]*

**Response 1:** We appreciate the reviewer's constructive comments and agree that detailed validation of simulations against observations are indeed essential for ensuring the reliability of the results. Given the inherent challenges in observing stratospheric ozone in the troposphere, assessing stratospheric intrusions and their vertical structure is particularly difficult. To address this concern, we have undertaken extensive validations and revisions to improve our paper as follows:

1. **Comparing the spatial structure of simulated SI ozone with Aura OMI data:** We compared the spatial distribution of simulated SI ozone with tropospheric ozone observations from the OMI remote sensing data (in Section 2.4). The vertical variations of OMI and SI ozone exhibit similar spatial patterns over CEC in the troposphere, while the simulated SI ozone concentrations are notably lower than those observed by OMI. This comparison underscores the reasonable relationships between stratospheric impact and total ozone (SI + tropospheric production) in the troposphere, thereby validating the reliability of our simulation methodology.

2. **Examining correlations with stratospheric ozone tracers:** We evaluated the correlation between simulated SI ozone and two stratospheric ozone tracer reanalysis datasets (EAC4 and CAM). This comparison, detailed in Section 2.4, shows a strong correlation, with coefficients in most sub-regions exceeding 0.7 and passing the confidence threshold of 99.9%.

3. **Comparison with ground-based observations:** We compared the SI events identified in our simulations with those identified from ground-based records. This analysis, presented in Section 3.3, demonstrates that our simulations achieve a high hit rate for SI events across nearly all sub-regions, indicating that our methodology effectively identifies SI events at the ground level.

4. We have removed the comparison with TOST data, as the TOST dataset was too coarse to be appropriate.

- The comparisons with OMI satellite data and stratospheric ozone tracer reanalysis datasets are detailed in Section 2.4 (Lines 210-247):

**2.4 Validation of the Lagrangian simulations**

[revised manuscript text omitted]